

# Neural network quantum states analysis of the Shastry-Sutherland model

**Matěj Mezera[1,2], Jana Menšíková[3,4], Pavel Baláž[3*] and Martin Žonda[1†]**

**1** Department of Condensed Matter Physics, Faculty of Mathematics and Physics,
Charles University, Ke Karlovu 5, Praha 2 CZ-121 16, Czech Republic
**2** Department of Mathematics and Computer Science, Freie Universität Berlin,
Arnimallee 12, 14195 Berlin, Germany
**3** FZU – Institute of Physics of the Czech Academy of Sciences,
Na Slovance 1999/2, 182 21 Prague 8, Czech Republic
**4** Institute of Theoretical Physics, Faculty of Mathematics and Physics,
Charles University, V Holešovičkách 747/2, 180 00 Praha 8, Czech Republic

⋆ balaz@fzu.cz , † martin.zonda@matfyz.cuni.cz

## Abstract

We utilize neural network quantum states (NQS) to investigate the ground state properties of the Heisenberg model on a Shastry-Sutherland lattice using the variational Monte Carlo method. We show that already relatively simple NQSs can be used to approximate the ground state of this model in its different phases and regimes. We first compare several types of NQSs with each other on small lattices and benchmark their variational energies against the exact diagonalization results. We argue that when precision, generality, and computational costs are taken into account, a good choice for addressing larger systems is a shallow restricted Boltzmann machine NQS. We then show that such NQS can describe the main phases of the model in zero magnetic field. Moreover, NQS based on a restricted Boltzmann machine correctly describes the intriguing plateaus forming in magnetization of the model as a function of increasing magnetic field.



# 1 Introduction

The neural network quantum states (NQSs) [1–10] have recently emerged as a promising alternative to common trial states in variational Monte Carlo (VMC) studies of quantum many-body problems, especially lattice spin models. This research is driven by the fact that neural networks (NNs) are universal function approximators [11] as well as by the astonishing progress in the field of machine learning (ML) in general. These advancements already led to a number of effective ML applications suitable for the basic research of quantum systems and technologies [12–15]. For example, even simple NQSs, such as the restricted Boltzmann machine (RBM), allow us to investigate the ground-state properties of various quantum spin models. It was already shown that RBM can outperform standard trial states in the variational search of the ground-state energies of the antiferromagnetic Heisenberg model [1]. Very promising results have also been obtained for frustrated spin systems, such as the $J_1-J_2$ model [16–19]. Here NQSs can be trained to capture the nontrivial sign structure of the ground state and in some cases have even achieved state-of-the-art accuracy [20] that delivers cutting edge results. Nevertheless, two-dimensional frustrated quantum spin models continue to be a challenge for NQSs as well as for other methods [21]. For example, it is not clear yet how to choose an optimal neural network architecture for a particular frustrated system, how important is the role of the trial state symmetries in the learning process, or if an NQS with favorable variational energy also encodes a physically correct state.

Not all of these issues are specific to NQSs. Results of any VMC calculations are dictated to a large extent by the properties and limitations of the trial states used. An inappropriately chosen variational state, that is, one with a small overlap with the ground state, can still give a good estimate of the ground-state energy [22]. If some additional information is known about the ground state, e.g., its symmetries, one can pick a more restrictive variational state function. However, this is often not an optimal strategy if the goal is to find new phases or to locate a phase boundary. In principle, NQSs could be a remedy for such problems. It is reasonable to expect that a single, but expressive enough, NQS can be used to approximate distinct phases. This assumption is supported by the results of Sharir et al. [23] who showed that NQSs can have even higher expressive power than matrix product states [24] and projected entangled pair states [25] as these can be efficiently mapped to a subset of NQSs. In other words, NQSs can be effectively utilized to a larger class of quantum states than these powerful formalisms

which are known primarily from their usage in Density Matrix Renormalization Group (DMRG) but are also utilized as variational states in VMC [1, 22, 26].

In practice, it is not yet clear how to achieve this in a general case. Despite tremendous progress, the research of frustrated quantum spin magnets is still in the stage of testing and developing NQS architectures for simple models, often focusing primarily on reaching the best variational energy in particular regimes [6, 16, 17, 19]. In the present work, we aim for a different target. We want to demonstrate that even shallow NQSs can be sufficient for the investigation of qualitatively different ground-state orderings including states forming only in a finite magnetic field. To this goal, we focus on the ground state of antiferromagnetic Heisenberg Hamiltonian on Shastry-Sutherland lattice known as the Shastry-Sutherland model (SSM) which we introduce in more detail in Sect. 2. To our knowledge, this model of frustrated quantum spin system has not been previously addressed within the NQS context, yet it seems to be an ideal testbed for our purposes.

SSM was already investigated by a number of methods, including exact diagonalization (ED) techniques [27–31], quantum Monte Carlo [32], various versions of DMRG [33–38], perturbation theory [39–42] and even quantum annealing [43]. These studies have shown that SSM has a rich ground-state phase diagram. In a zero magnetic field these include regions such as singlet spin dimer phase, antiferromagnetic Néel state, spin plaquette singlet phase and probably other phases. The introduction of a finite magnetic field further complicates the picture. Consequently, it is challenging to find a single variational function that can correctly approximate the whole ground-state phase diagram.

In addition, there are still open questions related to the ground-state phase diagram in zero as well as in the finite magnetic field, even in some experimentally relevant regimes of the model. This is important because several magnetic materials have a structure topologically equivalent to SSM. The most notable examples are $SrCu_2(BO_3)_2$, $BaNd_2ZnO_5$ and rare earth tetraborides $RB_4$ (R=Dy, Er, Tm, Tb, Ho) [44–48]. All exhibit an intriguing step-like dependence of the overall magnetization on the external magnetic field, which has been found to be inherent to SSM [49, 50]. Here, each plateau reflects a stable nontrivial spin ordering. The magnetic behavior of these materials is not yet fully understood. This together with other open problems, e.g., the prospect of a narrow spin liquid phase in a zero magnetic field, further motivates the investigation of SSM and its generalizations [42, 51–53].

Therefore, SSM presents a model system that has the right combination of properties that are well understood and can be used to benchmark various NQSs, and of open problems that can be potentially illuminated by these variational techniques. This includes the possibility to address the rather complex behavior of a system in relation to a changing magnetic field.

The present work consists of two main parts. In the first one we explore SSM by employing a number of NQS architectures and we test them against ED results for small lattices in zero magnetic field. Here the primarily goal is to find one or few networks that are able to capture the main well-understood ground-state orderings of SSM. Simultaneously, we require these NQSs to have a high chance to describe the magnetization plateaus as well. This means that the ideal network has to give a solid approximation of the ground-state orderings even when no conditions on the total magnetization are imposed. Consequently, we do not focus on getting the best possible variational energy for a particular set of parameters. Rather, we require a good approximation of the energy in distinct regimes of the model, a correct description of the particular orderings, and reasonable computational complexity that allows the usage of the NQS on larger lattices. We argue that when precision, generality, and computational costs are taken into account, a shallow RBM with complex parameters is still a good choice.

In the second part, we introduce a refined learning protocol for RBM NQS and test it for a wide range of model parameters and different network sizes. We then utilize this protocol in the study of larger systems. We first investigate the zero magnetic field scenario and demon-

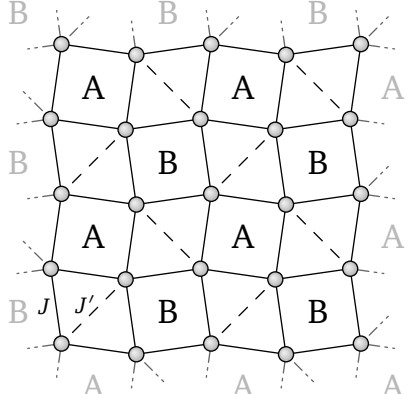

Figure 1: (a) The Shastry-Sutherland lattice. Bonds with coupling strength $J$ are represented by solid lines, while bonds with $J'$ by dashed ones. The letters A and B divide the "empty" squares into two subsets, which are used to define the plaquette order parameter.

strate that RBM is expressive enough to capture all main phases of the system. We then move to the model in a finite magnetic field and show that, with the right learning strategy, RBM is able to capture the magnetization plateaus crucial for the description of real materials. This opens a possibility that NQSs could be used to investigate several open problems, such as the existence of still opaque spin-liquid phase and other orderings predicted but not yet confirmed in SSM.

## 2 Shastry-Sutherland model

SSM is described by the Hamiltonian

$$\hat{H} = J \sum_{\langle i,j \rangle} \hat{\boldsymbol{S}}_i \cdot \hat{\boldsymbol{S}}_j + J' \sum_{\langle i,j \rangle'} \hat{\boldsymbol{S}}_i \cdot \hat{\boldsymbol{S}}_j - h \sum_i \hat{S}_i^z, \tag{1}$$

where $\hat{\boldsymbol{S}}_i = \frac{1}{2}\hat{\boldsymbol{\sigma}}_i$ is the spin-1/2 operator at the $i$-th site with $\hat{\boldsymbol{\sigma}}_i$ being the vector of Pauli matrices. The first term represents the exchange coupling between the nearest neighbors on a square lattice (solid lines in Fig. 1). The second term is a sum over specific diagonal bonds arranged in a checkerboard pattern (dashed lines in Fig. 1). Note that these sums are interpreted in terms of nodes, i.e., there is no double counting. Both coupling constants are antiferromagnetic ($J, J' > 0$) and we set $J'$ as the unit of energy in the whole paper. The last term describes the influence of the external magnetic field $h$ pointing to the $z$-direction.

### 2.1 Basic properties of the ground state

The basic structure of the SSM ground state phase diagram is well understood. As illustrated in Fig. 2, the SSM at $h = 0$ has at least three distinct ground-state orderings. These are the *dimer singlet* (DS) state for ($J' \gg J$), the *Néel antiferromagnetic* (AF) ordering ($J' \ll J$) and the *plaquette singlet* (PS) state in between. The phase transition from the DS to the PS state is of the first order [29], but the nature of the transition from PS to AF is still in debate. The ED study of Nakano and Sakai [30] suggests that the supposed PS phase actually consists of at least two distinct phases. In addition, some recent studies argue that there is a so-called *deconfined quantum critical point* (DQCP), which separates a line of first-order transitions or, potentially, a narrow gapless *spin liquid* (SL) phase [37, 38, 54].

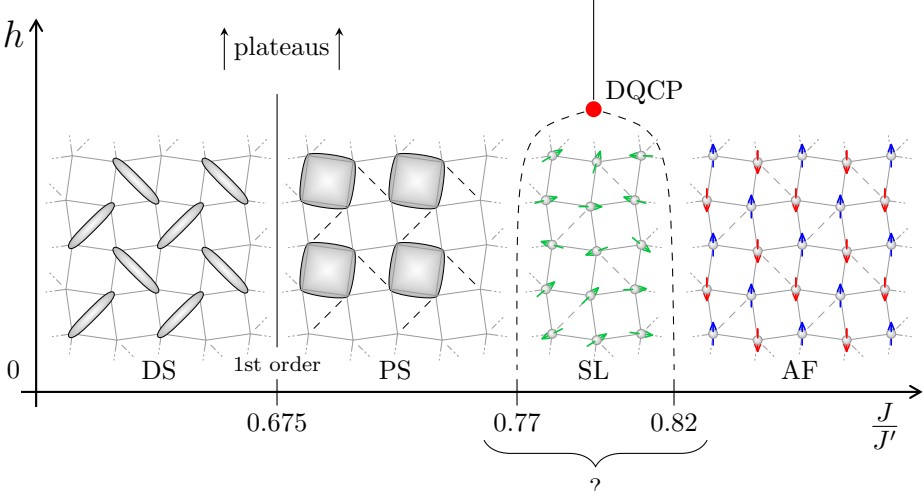

Figure 2: Illustration of the SSM phase diagram for small $h$ based on the results from Ref. [38]. There is a first-order transition at $J/J' \approx 0.675$ between DS and PS phases. The gray squares in PS depict the plaquette singlets. The nature of the transition between the PS and AF phases remains unresolved. It is not clear whether there is a narrow spin liquid phase, a DQCP or just a second order transition in the region labeled with a question mark.

Nevertheless, even without focusing on the possible DQCP and SL phase, the three main orderings, namely DS, PS, and AF, already pose a sufficient challenge for a single variational state because of their distinctive character and symmetries.

The **DS phase** is formed by an exactly (analytically) accessible state [55]. Numerous analytical and numerical methods have verified that it remains the ground state up to $J/J' \approx 0.675$ [29, 30, 37]. In the limiting case of $J \ll J'$, the system is equivalent to an ensemble of independent spin dimers, each of which forms a singlet ground state. The DS ground state is thus a direct product of dimer singlet states

$$|\psi\rangle_{\text{DS}} = \bigotimes_{\langle i,j \rangle'} \frac{1}{\sqrt{2}} \left( |\uparrow\downarrow\rangle_{i,j} - |\downarrow\uparrow\rangle_{i,j} \right). \tag{2}$$

As such, it is antisymmetric with respect to the exchange of two intradimer spins and symmetric with respect to transformations rearranging only the spin pairs without swapping the intradimer spins. The energy of the ground state of the dimer is

$$E_{\text{DS}} = -\frac{3}{4} J' N_{\text{D}}, \tag{3}$$

where $N_{\text{D}}$ is the number of dimers and $N_{\text{D}} = N/2$ for lattice with periodic boundary conditions.

The **PS phase** can be understood as weakly coupled plaquette singlet states illustrated in Fig. 2. Plaquette singlet is a ground state of an isolated 4-spin Heisenberg cluster with four bonds arranged in a cycle [29]. The pattern of the plaquette singlets in Fig. 2 indicates that the PS state is two-fold degenerate.

It is important to stress again that the relevant range $J/J'$ discussed here ($0.675 \lesssim J/J' \lesssim 0.82$) could be much more complex. As mentioned above, it has been argued that at $J/J' \approx 0.70$ the PS phase splits into two distinct regions with quantitatively different behaviors [30, 37, 38, 54]. For the sake of simplicity, we omit this possibility in most of our discussion. Nevertheless, this might be important for more detailed future studies.

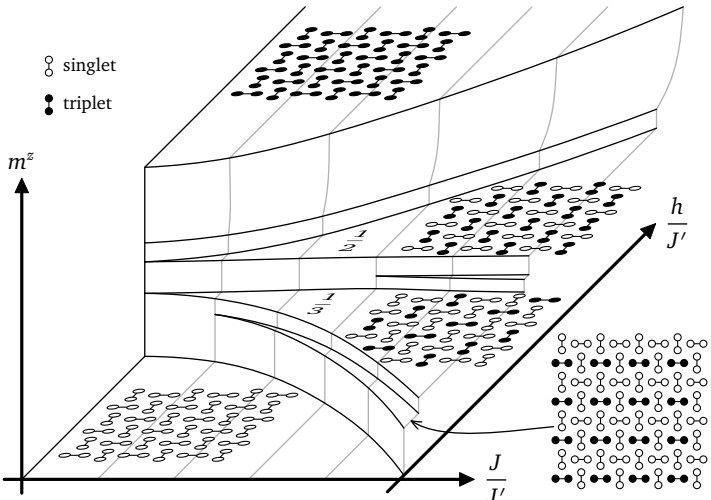

Figure 3: A simplified illustration of magnetization as a function of the external magnetic field $h$ and coupling constant $J$ inspired by Ref. [39]. A more detailed illustration would contain additional steps (e.g., supersolid phase); however, their actual position and width are not clear yet. Singlet and triplet arrangements are displayed for some of the plateaus (namely, $m^z = 1, 1/2, 1/3$ and $1/4$ plateaus are shown).

The **AF phase** stabilizes when $J/J' \gtrsim 0.82$. When $J'$ becomes negligible, the ground state of SSM is approaching the ground state of the antiferromagnetic Heisenberg model with only nearest-neighbor bonds on a square lattice. Although this state is not analytically accessible, it has previously been explored by Monte Carlo (MC) simulations [27]. Using the first-order correction to these quantum MC results, the energy of the SSM in the AF phase was estimated [27] to be

$$E_{\text{AF}} = (0.102J' - 0.669J)N \,, \tag{4}$$

where $N$ is assumed to be large.

A more detailed discussion of the symmetries of these three states is postponed to the Appendix C. Note that the three main phases DS, PS, and AF are reasonably understood, and simultaneously, they differ qualitatively. This is one of several qualities of the model that make the SSM a suitable testbed for NQSs.

So far we have discussed the $h = 0$ case. When we introduce a finite magnetic field to the DS phase in Eq. (2), some dimers can morph into triplet states. These triplets are formed in repeating patterns, e.g., checkerboard, stripes, or more complex configurations (for illustration, see Fig. 3), giving rise to stable plateaus of constant magnetization in increasing magnetic field.

Because each plateau signals a distinct stable ordering, it also presents a challenge for the NQSs. Particularly so because a finite magnetic field does not allow for a simple restriction of the Hilbert space to its zero magnetization part. This restriction was heavily utilized in previous NQS investigations of quantum spin models. Note that it is mostly these plateaus that make SSM interesting experimentally. Good examples are $SrCu_2(BO_3)_2$, $BaNd_2ZnO_5$, $CaCo_2Al_8$ and rare-earth tetraborides $RB_4$ (R=Dy, Er, Tm, Tb, Ho)) [44–48] which all exhibit the intriguing step-like dependence of the overall magnetization on the external magnetic field or show magnetic frustration and can be modeled by SSM or its generalizations.

# 3 Methods

## 3.1 Variational Monte Carlo and machine learning

VMC is a standard method that allows us to stochastically evaluate the expectation values of quantum operators without the need to probe the full Hilbert space. Suppose $\hat{H}$ is a fixed Hamiltonian operator and $|\psi_{\boldsymbol{\theta}}\rangle$ is a trial wave function that depends continuously on a set of parameters $\boldsymbol{\theta}$. VMC searches for a ground state of $\hat{H}$ or its approximation in a variational way. The goal is to minimize the variational energy

$$E_{\boldsymbol{\theta}} = \langle \hat{H} \rangle_{\boldsymbol{\theta}} := \frac{\langle \psi_{\boldsymbol{\theta}} | \hat{H} | \psi_{\boldsymbol{\theta}} \rangle}{\langle \psi_{\boldsymbol{\theta}} | \psi_{\boldsymbol{\theta}} \rangle} \geq E_0 \tag{5}$$

with respect to the vector of parameters $\boldsymbol{\theta}$, where $E_0$ is the true ground-state energy providing the lower energy bound. We utilize a fixed orthonormal basis $\{|\boldsymbol{\sigma}^z\rangle\}$ of the $z$-projected $\frac{1}{2}$-spins and use the following notation

$$|\psi_{\boldsymbol{\theta}}\rangle = \sum_{\boldsymbol{\sigma}^z} \psi_{\boldsymbol{\theta}}(\boldsymbol{\sigma}^z) |\boldsymbol{\sigma}^z\rangle, \quad \text{where} \quad \langle \boldsymbol{\sigma}^z | \psi_{\boldsymbol{\theta}} \rangle \equiv \psi_{\boldsymbol{\theta}}(\boldsymbol{\sigma}^z), \tag{6}$$

as is typical in NQS studies [56]. The variational energy in Eq. (5) is, in the jargon of ML, a *loss function*. Using this loss function, the parameters $\boldsymbol{\theta}$ are optimized to obtain the lowest energy state that the chosen variational function can represent. In our calculations, we use the VMC implementation from the NetKet NQS toolbox [9,56].

In general, the form of the trial wave function $\psi_{\boldsymbol{\theta}}(\boldsymbol{\sigma}^z)$ restricts the optimization process to a subset of the Hilbert space. An improper choice of the ansatz can bias the approximation towards a wrong phase or even can make the approach to the correct state impossible. Clearly, this is where one can expect that NQSs could outperform standard variational states due to their high expressiveness.

## 3.2 Neural network quantum states

Here, we explore several NQS architectures [6,9]. We chose these particular networks due to their successful application in previous studies of other Heisenberg models.

***Restricted Boltzmann machine (RBM)*** is a generative artificial NN constituted of a visible layer with $N$ nodes (one for each lattice site) fully connected with a single hidden layer with $M = \alpha N$ nodes (hidden degrees of freedom) where $\alpha$ is the *hidden layer density* [1]. It can be used to define an NQS

$$\log \psi_{\boldsymbol{\theta}}(\boldsymbol{\sigma}^z) = \sum_i \sigma_i^z a_i + \sum_j \log\left[ 2\cosh\left( \sum_i W_{ij}\sigma_i^z + b_j \right) \right], \tag{7}$$

where the vector $\boldsymbol{\theta}$ contains the variation network parameters $\boldsymbol{\theta} = \{\boldsymbol{a}, \boldsymbol{b}, \mathbf{W}\}$. This NQS can be interpreted as a one-layered fully-connected neural network with $\log\cosh$ activation function followed by a summation of the outputs and additional summation of visible biases [1]. Note that complex-valued parameters are necessary in order to represent generally complex-valued wave function outputs.

The size of the visible layer $N$ is fixed by the size of the investigated spin system. However, the expressive power of RBM can be modified by changing $\alpha$. The number of variation parameters of RBM is $\mathcal{O}(\alpha N^2)$.

***Modulus-phase split real-valued RBM (rRBM):*** Complex parameters, which generally make the learning process harder, can be avoided by introducing two independent real-valued

NNs [18, 57] to represent the modulus $A(\boldsymbol{\sigma}^z)$ and the phase $\Phi(\boldsymbol{\sigma}^z)$ of the wave function separately

$$\log\psi_{\boldsymbol{\theta}}(\boldsymbol{\sigma}^z) = A(\boldsymbol{\sigma}^z) + \mathrm{i}\Phi(\boldsymbol{\sigma}^z). \tag{8}$$

Unlike in the Ref. [57] where rRBM architecture proved to be advantageous in the investigation of transverse-field Ising model, we have experienced that for SSM, the rRBM shows worse results than complex-valued RBM. This is in accord with the recent study of other frustrated systems, namely the $J_1 - J_2$ model [19]. Consequently, we discuss the results of this network only briefly in Chapter 4.1 and focus predominately on complex-valued architectures.

**Symmetric variant of RBM (sRBM):** Carleo and Troyer [1] used translational symmetries to reduce the number of variational parameters in RBM. They replaced the fully connected layer with a convolutional layer and set the visible biases to the constant value $a^f$ across each convolutional filter $f$. The resulting expression for its output is

$$\log\psi_{\boldsymbol{\theta}}(\boldsymbol{\sigma}^z) = \sum_{f=1}^{F}\sum_{g\in G}\left\{a^f\underbrace{\sum_{i=1}^{N}T_g(\boldsymbol{\sigma}^z)_i}_{\sum_i^N \sigma_i^z = m^z} + \log\left[2\cosh\left(\sum_{i=1}^{N}w_i^f T_g(\boldsymbol{\sigma}^z)_i + b^f\right)\right]\right\}. \tag{9}$$

Here $\boldsymbol{T}_g$ denotes a symmetry transformation of a spin configuration according to an element $g$ from the symmetry group $G$ of order $|G|$. The index $f$ denotes different feature filters. The number of these filters $F$ determines the size of the network $M = F|G|$. The resulting sRBM has fewer variational parameters than the RBM by a factor of $|G|$. We can view this approach as binding the values of some of the $\mathcal{O}(\alpha N^2)$ parameter making the total asymptotic number of parameters $\mathcal{O}(\alpha N)$. Carleo and Troyer [1] also showed that this approach significantly improves the convergence and accuracy of the ground states of the antiferromagnetic Heisenberg model on a square lattice. However, this approach suffers from two crucial disadvantages in more general circumstances. The first drawback is that visible biases are inherently constant for each filter $f$ which significantly lowers the expressiveness of the network as discussed later in this section. As we show in Appendix B, the sRBM architecture cannot be modified to ease this condition while preserving symmetries. The second drawback is that sRBM is not applicable if the ground state does not transform under the trivial irreducible representation (irrep) of a given symmetry group.

To illustrate the problem, let us consider a single spin dimer (i.e., a single bond of SSM with $J = 0$, $J' = 1$ and $h = 0$). Its ground state is a singlet $|\psi_0\rangle = (|\uparrow\downarrow\rangle - |\uparrow\downarrow\rangle)/\sqrt{2}$. The symmetry group of the single-dimer Hamiltonian contains just two operations – an identity and a swap of both spins $G = \{g_{12}, g_{21}\}$. If we apply the swap operation to the ground state, we obtain $\hat{\boldsymbol{T}}_{g_{21}}|\psi_0\rangle = (|\downarrow\uparrow\rangle - |\downarrow\uparrow\rangle)/\sqrt{2} = -|\psi_0\rangle$. Although this state is a multiple of the ground state, we see that it does not transform under the trivial irrep because one of its characters is $\chi_{g_{21}} = -1$. Since sRBM represents only states with $\hat{T}_g|\psi\rangle = |\psi\rangle$; $\forall g \in G$, this symmetry should not be used in sRBM. Note that we do not strictly follow this rule and sometimes use all available lattice symmetries. The reason is that this leads to NQS with a small number of parameters that are easy to optimize. The resulting variational energy can then be compared with the energy obtained with RBM with the same $\alpha$ to check how well the full network is optimized, i.e., if it leads to lower energy than sRBM. If not, this signals that the variational energy of RBM can be lowered by better learning.

**Projected RBM (pRBM):** Recently, Nomura [58] introduced an alternative way to symmetrize RBM (or any other NN) using a quantum-number projection (also called *incomplete symmetrization operator*)

$$\psi_{\boldsymbol{\theta}}^G(\boldsymbol{\sigma}^z) = \sum_{g\in G}\chi_{g^{-1}}\psi_{\boldsymbol{\theta}}(\boldsymbol{T}_g(\boldsymbol{\sigma}^z)), \tag{10}$$

where $g$ is an element of the given symmetry group $G$ and $\chi_g$ is its character from the irrep in question. The wave function on the right-hand side may be arbitrary and it can be shown that the function on the left-hand side satisfies the desired transformation property $\psi_{\boldsymbol{\theta}}^G(T_g(\boldsymbol{\sigma}^z)) = \chi_g \psi_{\boldsymbol{\theta}}^G(\boldsymbol{\sigma}^z)$ in case of one-dimensional representation or $\psi_{\boldsymbol{\theta},a}^G(T_g(\boldsymbol{\sigma}^z)) = \sum_{b=1}^d D(g)_b^a \psi_{\boldsymbol{\theta},b}^G(\boldsymbol{\sigma}^z)$ for more-dimensional irreps, where functions $\psi_{\boldsymbol{\theta},a}^G$ form a basis of $d$-dimensional irrep $D(g)_b^a$. Unfortunately, pRBM makes the learning process of NN much more expensive than sRBM. The computational time increases by a factor of $|G|$ producing a computational cost $\mathcal{O}(\alpha N^2 |G|)$. On the other hand, pRBM implementation does not suffer from the problems mentioned for sRBM and it can be generalized by setting mutually independent visible biases (see Appendix B).

***Group-convolutional NN (GCNN)***: Group equivariant convolutional NNs represent a promising class of NNs built inherently on symmetries. They were proposed by Cohen and Ni [59] as a natural extension of the well-known convolutional neural networks. While convolutional networks preserve invariance under translations, GCNN are equivariant under the action of an arbitrary group $G$ (which may contain a subgroup of translations). Roth and MacDonald [60] further improved GCNNs so that they can transform under an arbitrary irreducible representation of $G$, which is more suitable for NQSs for SSM. GCNN can be composed of any number of hidden layers. The first and subsequent layers are given by

$$f_g^1 = f\left(\sum_{i=1}^N W_{g^{-1}i}^0 \sigma_i^z + b^0\right), \quad f_g^{k+1} = f\left(\sum_{h\in G} W_{g^{-1}h}^k f_h^k + b^k\right), \tag{11}$$

where $f$ is a nonlinear activation function (the output is typically a vector since GCNN can have multiple parallel feature filters) and $f_g^1$ is a 1st-layer feature vector corresponding to group element $g$. The result of the last layer $f_g^K = f_g^{(j)K}$, where $(j)$ denotes the individual features of the layer, is then projected in a fashion similar to that of pRBM

$$\psi(\boldsymbol{\sigma}^z) = \sum_{g\in G}\sum_j \chi_{g^{-1}} \exp\left(f_g^{(j)K}\right). \tag{12}$$

The main advantage over symmetrizing an arbitrary deep network by the formula form Eq. (10) is that we do not need to evaluate the forward pass of the nonsymmetric wave function $|G|$ times. This is achieved because each layer of the GCNN fulfills *equivariance*. GCNN with $K$ layers and a typical number of feature filters $F$ in each layer has $\mathcal{O}(FN + KF^2|G|)$ parameters.

***Jastrow network:*** As a baseline, we also use a Jastrow network based on the standard Jastrow ansatz [61, 62]

$$\psi_{\boldsymbol{\theta}} = \exp\left(\sum_{i,j} \sigma_i^z W_{i,j} \sigma_j^z\right), \tag{13}$$

where the variational parameters $\boldsymbol{\theta} = \{W_{i,j}\}$ form a matrix of size $N \times N$. The Jastrow ansatz is physically motivated by two-body interactions and assigns trainable parameters $W_{i,j}$ to pairwise spin correlations. The number of its parameters scales as $\mathcal{O}(N^2)$.

The complicated sign structure of the complex phases of the basis coefficients that form the ground-state wave function presents a major challenge in optimizing the parameters of a variational function of a frustrated spin system. In case of Heisenberg model on a bipartite lattice consisting of sublattices $\mathcal{A}$ and $\mathcal{B}$ (i.e., SSM with $J' = 0$), this can be solved using the Marshal sign rule (MSR) [63]. The MSR states that the sign of $\psi(\sigma^z)$ is given by $(-1)^{N_{\mathcal{A}}^{\uparrow}(\sigma^z)}$ where $N_{\mathcal{A}}^{\uparrow}(\sigma^z)$ is the total number of up-spins on a sublattice $\mathcal{A}$. Because this alternates with a spin-flip, it can be difficult for NN to learn the correct signs. However, it is possible to circumvent this problem in two analogous ways.

If the sign structure is dictated by MSR, the Hamiltonian can be gauge transformed by changing the signs of some terms to make all wave function coefficients positive in the transformed basis. In particular, we change $\sigma^x \to -\sigma^x$ and $\sigma^y \to -\sigma^y$; for $\forall \sigma \in \mathcal{A}$. The same result can be also obtained by setting the visible biases to $a_i = i\pi/2$ for $i \in \mathcal{A}$ and $a_i = 0$ for $i \in \mathcal{B}$ as this exactly reconstructs the Marshall sign factor (up to an overall constant factor). In other words, the biases can be set to play the role of a Marshall basis. What is important here is that in the general case the simple Marshall sign rule is not always applicable. Especially problematic are systems with strong frustration [18, 19, 64]. The advantage of using the visible biases instead is that their setting does not have to be known ahead, as it can be, despite possible technical difficulties, learned. Therefore, it is beneficial to include visible biases whenever allowed by the architecture. An additional bonus is that free visible biases also allow one to overcome an improper initialization of weights.

## 4  Results

### 4.1  Comparison of different NQSs architectures

It is too expensive to apply all NQSs introduced above to investigate the ground-state phase diagram of SSM at large lattices. Therefore, in the first part of our investigation, we benchmark these NQSs against the exact results on smaller lattices obtained by the Lanczos ED method. The aim is to identify a network that is both expressive enough to cover various phases and computationally tractable even for large lattices. We focus on a regular lattice with $N = 4 \times 4 = 16$ points and an irregular lattice with $N = 20$ (see Appendix A). Throughout this paper, we apply periodic boundary conditions for all lattices used, unless explicitly stated otherwise. The irregular $N = 20$ is considered because $N = 16$ lattice has some undesirable properties, e.g., some extra symmetries with trivial irrep which favor symmetric networks. It also suffers from stronger finite-size effects and does not exhibit the PS phase. On the other hand, it is regular and easy to calculate.

We initially focus on the cases represented by $J/J' = 0.2$ (DS phase), $J/J' = 0.9$ (AF phase), and $J/J' = 0.63$. Case $J/J' = 0.63$ was chosen because it represents a realistic case, namely, it is the exchange parameter ratio for $SrCu_2(BO_3)_2$ at ambient pressure [34]. However, because its results are qualitatively in agreement with the case $J/J' = 0.2$, we discuss them together as the DS results. Note that we investigate the model with and without MSR. Since the goal here is to compare different networks, we estimate the accuracy of each architecture by comparing the average energy of the last 50 learning iterations $E_{50}$ with the exact result $E_{ex}$. Note that this means that we are not using just the lowest obtained energies but also test stability of the learning method. Consequently, the value of $E_{50}$ is typically greater than zero even when the network is able to reproduce the state exactly. The same computational protocol is used for each architecture. In particular, we used 2000 MC samples[1] and 1000 training iterations for three values of fixed learning rates $(0.2, 0.05, 0.01)$. Each particular combination of architecture and basis (MSR or direct) was computed four times for each learning rate (yielding 12 independent runs for each case of interest). This is to eliminate occasional events when NN gets stuck in a local energy minimum too far from the ground state. Zero magnetization was not implicitly assumed (i.e., we used local single-spin-flip Metropolis updates in VMC). We summarize our results in Table 1 where the values are $\min_i \frac{|E_{50}^i - E_{ex}|}{E_{ex}}$, with $i$ enumerating the twelve independent runs.

There are several results in Table 1 which were important for our decision on which network should be used in the detailed study of the phase diagram in larger lattices. Starting

---

[1]For $N = 16$, exact samples were used. For details, see ExactSampler in [9].

Table 1: Comparison of the precision (lower is better) of NQS variational results on lattices $N = 16$ and $N = 20$. The listed values were calculated as $\min_i \frac{|E_{50}^i - E_{\text{ex}}|}{E_{\text{ex}}}$, where $E_{50}^i$ is the average energy of the last 50 iterations of the $i$-th run. A number of variational parameters are also shown for each architecture. The difference between GCNN and GCNNt is that for GCNN we used all the symmetries and the correct characters for the expected ground state, whereas GCNNt utilized only the translation symmetry. The error 0.0 here means a relative error less than $10^{-7}$ which we consider as a "numerical precision" due to the standard MC errors which are typically larger even for $L = 16$.

| $N = 4 \times 4$ architecture | params | $J/J' = 0.2$ (DS) direct | MSR | $J/J' = 0.63$ direct | $J/J' = 0.9$ (AF) direct | MSR |
|---|---|---|---|---|---|---|
| Jastrow | 256 | $5.8 \times 10^{-5}$ | $1.1 \times 10^{-5}$ | $6.3 \times 10^{-6}$ | $6.2 \times 10^{-2}$ | $2.1 \times 10^{-2}$ |
| RBM ($\alpha = 2$) | 560 | $1.9 \times 10^{-5}$ | $1.6 \times 10^{-5}$ | $1.7 \times 10^{-5}$ | $4.7 \times 10^{-3}$ | $5.1 \times 10^{-3}$ |
| RBM ($\alpha = 16$) | 4368 | $2.8 \times 10^{-5}$ | $1.4 \times 10^{-5}$ | $1.9 \times 10^{-5}$ | $1.6 \times 10^{-3}$ | $1.1 \times 10^{-3}$ |
| rRBM ($\alpha = 2$) | 1088 | $2.3 \times 10^{-4}$ | $2.1 \times 10^{-4}$ | $5.2 \times 10^{-5}$ | $8.3 \times 10^{-3}$ | $7.9 \times 10^{-3}$ |
| rRBM ($\alpha = 8$) | 4352 | $2.3 \times 10^{-4}$ | $2.2 \times 10^{-4}$ | $1.1 \times 10^{-5}$ | $9.3 \times 10^{-3}$ | $7.8 \times 10^{-3}$ |
| sRBM ($\alpha = 4$) | 18 | 0.0 | 0.0 | $3.8 \times 10^{-6}$ | $4.8 \times 10^{-3}$ | $2.5 \times 10^{-3}$ |
| sRBM ($\alpha = 16$) | 69 | 0.0 | 0.0 | $2.7 \times 10^{-6}$ | $8.9 \times 10^{-4}$ | $3.8 \times 10^{-4}$ |
| sRBM ($\alpha = 128$) | 545 | 0.0 | 0.0 | $4.9 \times 10^{-6}$ | $1.3 \times 10^{-3}$ | $8.5 \times 10^{-5}$ |
| pRBM ($\alpha = 0.5$) | 136 | $7.9 \times 10^{-5}$ | $1.2 \times 10^{-4}$ | $6.9 \times 10^{-5}$ | $1.5 \times 10^{-3}$ | $8.5 \times 10^{-4}$ |
| pRBM ($\alpha = 2$) | 544 | $9.1 \times 10^{-6}$ | $2.2 \times 10^{-5}$ | $6.5 \times 10^{-6}$ | $7.1 \times 10^{-5}$ | $2.0 \times 10^{-5}$ |
| GCNN | 2188 | $6.2 \times 10^{-6}$ | $3.6 \times 10^{-6}$ | $1.2 \times 10^{-5}$ | $3.2 \times 10^{-5}$ | $3.6 \times 10^{-5}$ |
| GCNNt | 268 | $4.2 \times 10^{-7}$ | $5.1 \times 10^{-7}$ | $4.0 \times 10^{-7}$ | $4.9 \times 10^{-3}$ | $4.7 \times 10^{-3}$ |
| $N = 20$ | | $J/J' = 0.2$ (DS) | | $J/J' = 0.63$ | $J/J' = 0.9$ (AF) | |
| Jastrow | 400 | $1.1 \times 10^{-5}$ | $1.0 \times 10^{-3}$ | $2.5 \times 10^{-3}$ | $1.4 \times 10^{-1}$ | $3.0 \times 10^{-2}$ |
| RBM ($\alpha = 2$) | 860 | $2.2 \times 10^{-5}$ | $1.4 \times 10^{-5}$ | $7.5 \times 10^{-4}$ | $6.6 \times 10^{-3}$ | $6.2 \times 10^{-3}$ |
| RBM ($\alpha = 8$) | 3380 | $1.2 \times 10^{-5}$ | $1.7 \times 10^{-5}$ | $1.7 \times 10^{-3}$ | $2.2 \times 10^{-3}$ | $2.1 \times 10^{-3}$ |
| sRBM ($\alpha = 4$) | 85 | $1.2 \times 10^{-1}$ | $1.5 \times 10^{-1}$ | $1.4 \times 10^{-1}$ | $5.0 \times 10^{-2}$ | $1.4 \times 10^{-3}$ |
| pRBM (for AF) | 336 | $2.3 \times 10^{-1}$ | $2.3 \times 10^{-1}$ | $5.2 \times 10^{-2}$ | $3.5 \times 10^{-3}$ | $3.0 \times 10^{-3}$ |
| pRBM (for DS) | 336 | $7.1 \times 10^{-4}$ | $6.8 \times 10^{-5}$ | $1.2 \times 10^{-3}$ | $4.4 \times 10^{-2}$ | $4.7 \times 10^{-2}$ |

with RBM, one can see that networks with $\alpha = 2$ (560 parameters for $N = 16$ and 860 for $N = 20$) and 8 (3380 parameters for $N = 20$) and 16 (4398 parameters for $N = 16$) show similar precision, where the significantly larger networks are notably better (approximately three times) only in the AF phase. For the general case, considering the computational costs, this favors the computationally less demanding network with $\alpha = 2$. Also interesting is the comparison with the Jastrow network. Both architectures have comparable precision in the DS phase for $N = 16$, however, in the AF phase and in the DS phase for $N = 20$ with MSR, RBM is one or even two orders of magnitude more precise than the Jastrow ansatz.

For $N = 16$, the sRBM architecture demonstrates superior performance. The full automorphism group of the finite lattice has been used in its implementation. Despite the resulting small number of variational parameters, it shows excellent precision. In fact, a significant increase of $\alpha$ is not that advantageous (compare the cases $\alpha = 4$ and $\alpha = 128$). In the DS phase, the use of symmetries allowed sRBM to find the ground-state energies within the numerical precision (hence the zero error). Since sRBM can be thought of as RBM with additional constraints on the values of the weights, this already suggests that the learning protocol for

RBM can be improved, which we demonstrate in the next section. However, it is important to stress that the excellent results are a consequence of the special symmetries of the $N = 16$ lattice. Both the DS and AF states transform under the trivial irreducible representation, and the automorphism group is therefore applicable without special treatment. This is not true for the DS ground state in different tiles, including regular ones such as $N = 6 \times 6$ (for a more detailed discussion of the symmetries, see Appendix C). This is illustrated in the second part of Table 1 where sRBM with $\alpha = 4$ gives very poor results in the DS phase of $N = 20$ due to the improper treatment of symmetries. In short, using symmetries in sRBM for states that do not transform under a trivial irrep can make the variational energy significantly worse than for simple RBM. For $N = 20$, sRBM also fails in the AF phase, but only when adopting a direct basis. This implies that sRBM has trouble learning the correct sign structure of the state for larger lattices, which can be attributed to the fixed visible biases.

The remaining architectures, namely pRBM and GCNN, show excellent accuracy for $N = 16$. They clearly outperform all other networks in the AF phase. However, the results at $N = 20$ are less convincing, especially when one takes into account that these networks are more computationally demanding than RBM even for cases when RBM contains more parameters. Furthermore, the precision reached required the usage of correct symmetries of the expected state, i.e., the proper line form Table 2 in Appendix C. If one uses an improper one, i.e., if different state is expected, as illustrated by the last two lines in Table 1, the precision can drop by several orders of magnitude. Similarly, precision decreases significantly for both GCNN and pRBM when we use only the group of translations instead of the full symmetry group, as illustrated by GCNNt in Table 1. Note that for this case, the precision in the AF phase drops to the level of a simple RBM with $\alpha = 2$. The network is much better in the DS phase, but in the following chapter, we will demonstrate that even RBM with $\alpha = 2$ and modified learning protocol can reach the numerical precision in this phase. Although we cannot exclude that much better results could be obtained for the symmetrized pRBM and GCNN networks with a different learning protocol, considering their much higher computational demands and the necessity to identify a priori the correct irrep symmetries for each lattice type to make the learning efficient, the presented results favor RBM for the study of larger clusters.

The last question to be addressed here is whether using MSR would be beneficial. Table 1 shows several cases where MSR is favorable in the AF phase (e.g., for sRBM and $N = 16$), but this is not a general rule. In addition, its usage comes with a price as well. We have noticed that the MSR basis seems to strongly favor the AF ordering even for $J/J'$ where PS is already the ground state in exact results. We will discuss this briefly when addressing larger lattices.

To wrap it up, in general, the usage of MSR basis does not lead to significantly better results. With some exceptions, the networks presented here are able to approximate the ground-state energy quite well even without MSR. Therefore, we will mostly omit the MSR from further discussion. Furthermore, if the symmetry of the ground state is known, it is worth using this information in building the NN. If not, then the usage of just translations does not lead to a significant improvement of the precision. Fortunately, the complex-valued RBM with visible biases can give a very good approximation of the ground-state energy without any restrictions. Its clear advantage is that no preliminary information about the ground-state properties is needed. As such, it is suitable for problems where the character of the ground state or position of the phase boundary is unknown. In addition, the precision of RBM for SSM can be significantly improved using a different learning strategy discussed in the following section.

## 4.2 Investigation of the ground-state phase diagrams

Focusing solely on RBM allowed us to test several learning strategies and employ more precise MC calculations. What follows is a description of the best learning protocol we have found,

which we used to produce all the results discussed below. It proved to be beneficial to use more precise MC calculations already during training. We typically generate 4000–12000 MC samples at every sampling step. It was also more advantageous to run 10–30 independent learnings (with random initial variational parameters) with shorter learning times than to use few runs with a lot of learning iterations. We used approximately 2000 training iterations in each run. During learning, we have been lowering the learning rate $\eta$ by several discrete steps. Typically, we started with $\eta = 0.08$ ($\approx$200 iterations), then changed it to $\eta = 0.04$ ($\approx$1600 iterations), followed by $\eta = 0.02$ ($\approx$100 iterations), $\eta = 0.01$ ($\approx$100 iterations) and $\eta = 0.003$ ($\approx$50 iterations). The trained RBM was then used to calculate the expectation values of the energy and order parameters, introduced in the next section, where we used 12–60 thousand evaluation steps. Consequently, the Monte Carlo error bars in all the figures presented are negligible for small lattices. The relevant absolute error comes from the learning process or limitations of the NQS used. The state with the lowest energy (evaluated more precisely after training) of all independent runs was kept as the final result in the following discussion. Due to the stochastic fluctuations in the learned parameters, it was for some cases advantageous to refine the results by fine-tuning the final state multiple times with a high number of MC samples but a small number (5-10) of iterations and a small learning rate ($\eta \leq 0.001$) keeping the result with the lowest energy. Moreover, transfer learning was employed in some problematic regimes, as described below.

### 4.2.1 Ground-state orderings

As already discussed, good agreement of the variational energy with the exact one does not guarantee that the variational state correctly captures the character of the exact ground state, i.e., that it reflects the correct phase. To examine this and with the aim to see if RBM NQS can correctly describe the transitions between the phases, we calculate the order parameters for the three main expected orderings. They are constructed to be large (close to one) whenever the state is in the respective phase and small in other domains.

In particular, we define the order parameter for the DS phase as

$$\mathcal{P}_{\text{DS}} = -\frac{4}{3N} \sum_{\langle i,j \rangle'} \langle \hat{\boldsymbol{s}}_i \cdot \hat{\boldsymbol{s}}_j \rangle, \tag{14}$$

which reflects the fact that operator $\hat{\boldsymbol{S}}_1 \cdot \hat{\boldsymbol{S}}_2$ has for isolated dimer the expectation value $-\frac{3}{4}$ (singlet state). Therefore, $\mathcal{P}_{\text{DS}}$ is one in the DS phase and strictly lower in other phases.

For the PS order parameter, we use a definition based on order parameter from Ref. [38]

$$\mathcal{P}_{\text{PS}} = \frac{1}{\bar{N}} \left| \left\langle \sum_{r \in A} \hat{Q}_r - \sum_{r \in B} \hat{Q}_r \right\rangle \right|, \tag{15}$$

where the order parameter is given by the difference $\hat{Q}_r = \frac{1}{2}\left(\hat{P}_r + \hat{P}_r^{-1}\right)$, with $\hat{P}_r$ being the permutation operator. This operator performs a cyclic permutation of four spins on a plaquette (a square on the lattice without the diagonal bond $J'$) at position $\boldsymbol{r}$. Here, the first sum in Eq. (15) runs over the subset of squares A (see Fig. 1) and the second sum runs over the subset B. The meaning of this construction can be understood by looking at Fig. 2. Note that in the investigation of the plaquete ordering we utilized in addition to periodic boundary conditions (torus geometry) also a lattice with mixed ones. For periodic boundary conditions, we have $\bar{N} = N/4$ as all squares are used. For mixed ones, we followed Ref. [38] and use regular lattices with open boundary conditions in the $x$-direction with $L_x = 2L$ and periodic in the $y$-direction with $L_y = L$ so that $N = 2L^2$. However, the order parameter is calculated only in the central $L \times L$ square to mitigate the boundary effects. Hence, $\bar{N} = L^2/4$. The operator $\hat{Q}_r$

gives a large mean value in the plaquette singlet (gray square) and a value close to zero in the empty square between four plaquette singlets. For periodic lattices, we do not know which set of squares will become singlets, as the state is degenerate, therefore, we use the absolute value.

For the AF phase we employ the standard structure factor

$$\mathcal{P}_{\text{AF}} = \frac{1}{N^2} \sum_{ij} e^{i\boldsymbol{q}\cdot\boldsymbol{r}_{ij}} \left\langle \hat{\boldsymbol{S}}_i \cdot \hat{\boldsymbol{S}}_j \right\rangle, \tag{16}$$

where $\boldsymbol{r}_{ij}$ denotes the difference in discrete coordinates of spin $i$ and $j$, and we take $\boldsymbol{q} = (\pi, \pi)$ which measures the antiferromagnetic checkerboard ordering. Finally, in the case of finite magnetic field we use the normalized magnetization in the $z$-direction

$$\mathcal{M} = \frac{2}{N} \sum_i \left\langle \hat{\boldsymbol{s}}_i^z \right\rangle \tag{17}$$

to identify the expected plateaus in the magnetization. These expectation values are calculated using VMC for trained RBM NQS.

### 4.2.2 Zero magnetic field

We first investigate the phases of SSM in a zero magnetic field. Unlike the procedure used to compare different network architectures, here we restrict the Hilbert space by the condition $\mathcal{M} = 0$. Before moving to larger lattices, we test the RBM for $N = 20$ in a wide range of $J/J'$. We use the irregular lattice $N = 20$ because it shows an onset of the PS ordering (see the black dashed line in Fig. 4(a)) not present for smaller regular lattices. We also readdress the role of the parameter $\alpha$ within the new learning protocol, but start our discussion with the case $\alpha = 2$.

As is clear from the comparison of the ground-state energies in panels Fig. 4(b) and Fig. 4(c), the RBM variational energy agrees very well with the ED. The updated learning protocol ensures that the relative error in the $J/J' < 0.68$ region, i.e., for the DS phase, is on the order of the numerical precision already for $\alpha = 2$ despite not using any symmetries except for the condition $\mathcal{M} = 0$. The largest error is in the vicinity of the expected first-order phase transition from the DS to PS phases, but only from the side of the expected PS phase. Nevertheless, even here, the largest observed relative error in energy was approximately 1% for $\alpha = 2$.

Given the focus of our study, even more important than the energy error is the nature of optimized variational states. Panel (a) in Fig. 4 shows that a shallow network, i.e., RBM with complex parameters and $\alpha = 2$ is expressive enough to correctly capture the formation of the distinct DS (blue diamonds) and AF ordering (red crosses), as well as the onset of the PS phase (black circles). The agreement is far from perfect, though. Consistent with the results for the energy, the largest differences in order parameter values between RBM and ED are in the right vicinity of the expected phase transition. Here an error of 1% and less in the estimation of the ground-state energy translates into an error of tens of percents in the order parameters. Still, even here the RBM gives a correct qualitative picture. The position of the abrupt change of phase matches the exact result and there is a clear onset of the PS ordering. With increasing $J/J'$, the RBM results align again with the exact ones.

This benchmark shows that RBM with $\alpha = 2$ can easily capture the correct state in the DS phase, but gives worse results above the critical $J/J' \approx 0.68$. What is not clear is if the relative errors in panel (c) represent some inherent limitation of the RBM with small $\alpha$, e.g., a difficulty to set the correct sign structure of the frustrated state, or are related to the learning process. Gradually increasing $\alpha$ from 2 (blue circles) to 4 (red pluses), 8 (green pluses) and 16 (black

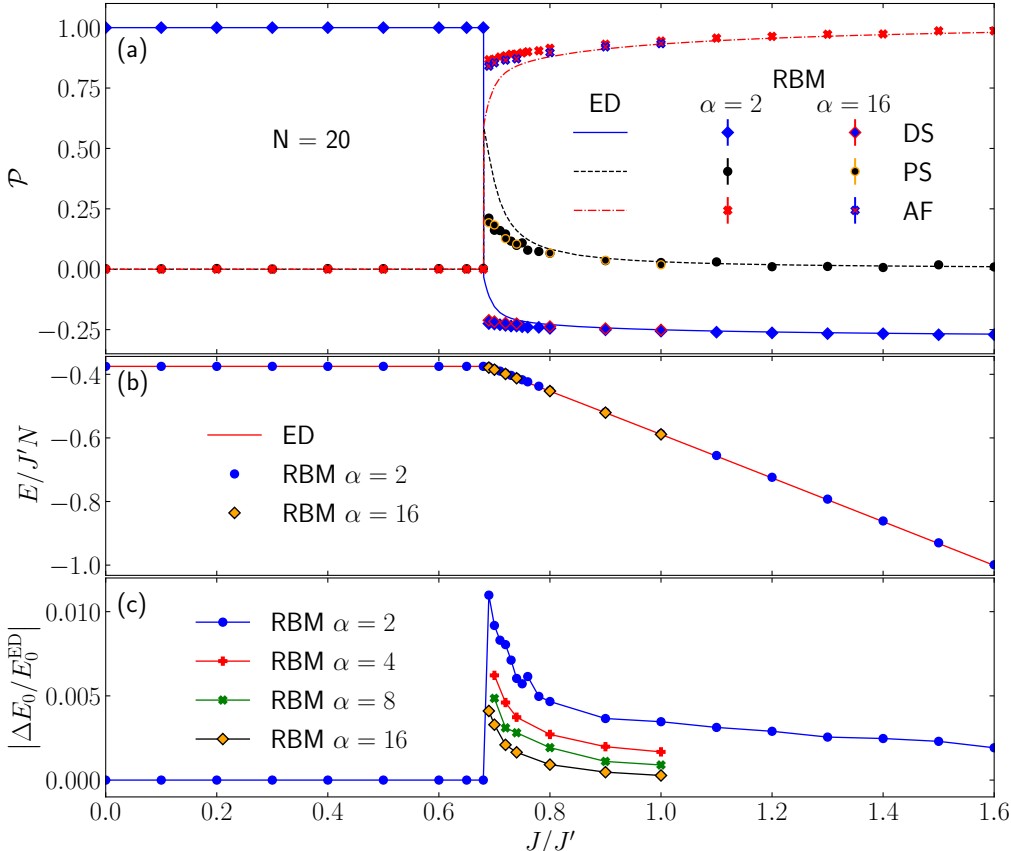

Figure 4: Comparison of exact (lines) and various RBM variational results (symbols) at irregular lattice $N = 20$. (a) Evolution of the order parameters. Here blue solid line (ED), pure blue diamonds (RBM with $\alpha = 2$) and blue diamonds with red edge (RBM with $\alpha = 16$) show the DS order parameter; black dashed line (ED), black circles (RBM with $\alpha = 2$) and black circles with yellow edge (RBM with $\alpha = 16$) show the PS order parameter; and red dot-dashed line (ED), red crosses (RBM with $\alpha = 2$) and red crosses with blue edge (RBM with $\alpha = 16$) show the AF order parameter. The results of symmetric variants of RBM are not shown, as they were comparable to the results presented for $J/J' > 0.68$ and well off the exact results for $J/J' \leq 0.68$. (b) The exact (red line) and RBM $\alpha = 2, 16$ ground-state energies. (c) Relative error in ground-state energy for the RBM with $\alpha = 2$ (blue circles), 4 (red pluses), 8 (green crosses) and 16 (black-yellow diamonds). Note that the relative error in the DS phase for RBM $\alpha = 2$ is at the level of numerical precision.

diamonds with yellow cores) in the problematic region lowers the relative error in energy. However, this significant improvement in energy leads only to a small improvement for the order parameters near the critical point. This is shown in panel (a) where the results calculated with RBM with $\alpha = 16$ are marked with the same symbols as for $\alpha = 2$ but highlighted via differently colored edges.

Using symmetric NQS symmetries did not significantly improve the results. We have tested the sRBM architecture with $\alpha = 4$ in direct as well as MSR basis using the same protocol as for RBM. The sRBM results have been comparable to RBM for $J/J' > 0.68$ and much worse than the RBM results below this critical value. This suggests that the issue is not entirely due to insufficient learning. On the other hand, the learning was the most difficult in the vicinity of the observed discontinuity. A significant fraction (often more than half) of the independent

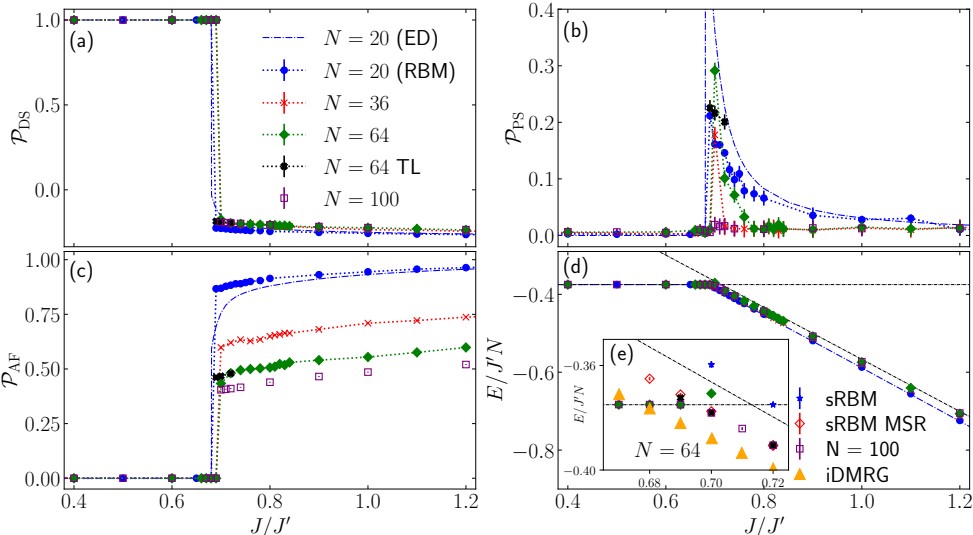

Figure 5: Evolution of order parameters for DS (a), PS (b), AF (c) and variational energy (d) as a function of $J/J'$ for $h = 0$ and various lattice sizes. All results in panels (a)-(d) have been obtained using RBM NQS with $\alpha = 2$ and VMC with exchange updates (simultaneous flip of two opposite spins in the basis state) for the Hilbert subspace restricted to $\mathcal{M} = 0$. The black dashed lines in panels (d) and (e) show the asymptotic energies for the DS (horizontal) and AF phase (tilted). The black crosses represent the results with $N = 64$ for which we have utilized transfer learning. The inset (e) shows the details of the variational energy for $N = 64$ in the vicinity of the phase transition calculated using RBM (green diamonds), sRBM in direct base (blue stars), sRBM with MSR (red empty diamonds) and three points calculated with RBM utilizing transfer learning (black crosses). The empty purple squares show the RBM results for $N = 100$, and the orange triangles are infinite DMRG results taken graphically from Ref. [37].

runs for $0.69 \leq J/J' \leq 0.72$ ended either in the wrong phase (DS) or even in a state with an energy much higher than the real ground state. This was not true for the rest of the $J/J'$ interval, where most of the independent runs with the same $\alpha$ showed very similar variational energies. Furthermore, the relative errors for all investigated RBM variants (including those not presented here) follow the same pattern. They are maximal just above the critical point and then, if we neglect some noise, they monotonically decrease with increasing $J/J'$. Yet, increasing $\alpha$ significantly lowers the variational energy even for $J/J' > 0.74$. This again suggests that the problem is indeed small $\alpha$. Ultimately, both statements seem to be correct. Significantly larger $\alpha$ than $\alpha = 16$ is needed to capture the critical region together with high-precision learning, that is, many independent runs.

After testing the RBM on small lattices and understanding its strength and limitations, we can now approach larger ones. We focus on $\alpha = 2$ as the increase in the precision of the variational energy obtained with larger $\alpha$'s does not significantly improve the estimates of the order parameters. Although we can not easily compare the VMC results with the exact diagonalization for larger lattices, we can use the exact asymptotic results for the energy in DS Eq. (3) and AF phase Eq. (4) to guide us.

Fig. 5 shows the evolution of the order parameters and energy for $N = 20, 36, 64$ and $N = 100$. The results agree very well with the exact result in the assumed DS phase and are between the exact energy of $N = 20$ and the asymptotic energy for large $N$ in the supposed *AF* phase up to several points in a very narrow region near the discontinuous phase transition discussed later.

Fig. 5 illustrates the usability of RBM for larger clusters. The presented results support the overall picture of the DS and AF phase separated by a narrow PS or at least its indication. Nevertheless, a much more thorough finite-size analysis would be necessary to assess the phase boundaries. For example, $\mathcal{P}_{AF}$ decreases with increasing system size in the whole relevant range of $J'$ which is in agreement with previous studies, e.g. [38]. Consequently, a careful and precise extrapolation of $\mathcal{P}_{AF}$ to the thermodynamic limit is needed to identify $J$ above which the AF ordering prevails. However, even in this respect, there is an issue. The point of the discontinuous phase transition from the DS phase to the PS phase should be $J/J' \simeq 0.675$, but our results at larger lattices push it to $J/J' \simeq 0.7$. Besides finite-size effects, this could also be related to two technical problems. The first is the difficulty of training the NQS in the vicinity of the discontinuous phase transition. The second is the tendency of the direct base to prefer DS over AF ordering. Both these issues can be seen in panel (e) (inset of panel (d)) with details of the $N = 64$ (and $N = 100$) results. Here, green diamonds show the RBM data, red empty diamonds are sRBM data with MSR basis, and blue stars are sRBM data for direct basis, all with $\alpha = 2$ for $N = 64$. Clearly, all these networks show (different) problems around the expected point of the phase transition. For $J/J' = 0.7$ and 0.72 sRBM with MSR gives energy lower than RBM and even lower than the energy of DS ordering. Therefore, the sharp transition must be placed below $J/J' = 0.7$. However, sRBM with MSR cannot correctly capture the onset of DS ordering. The sRBM network with direct basis illustrates the opposite problem. It overestimates the stability of the DS ordering.

Investigation of sRBM showed that the RBM results at $J/J' = 0.7$ are not yet fully converged. Because we have not been able to solve this problem using the direct approach, we utilized transfer learning. We used the RBM parameters trained for $J/J' = 0.74$ as a starting point to train the network at $J/J' = 0.72$, then used these results as a starting point for $J/J' = 0.70$, and finally these results for 0.69. That way we obtained lower variational energies for $J/J' = 0.72$ and 0.70 than in the direct approach or in the sRBM results, and the $J/J' = 0.72$ result dropped even below the DS energy. Interestingly, this also leads to an observable change in the order parameters (black crosses in all panels). In contrast to the $N = 20$ case, the PS ordering is especially sensitive to this change, as seen from the comparison of black crosses and green diamonds in panel (d). Even if it is suggested by the order parameters, the transfer learning technique has not reached the point of the expected phase transition below $J/J' = 0.69$. The reason is that the energy obtained at this point exceeds the DS energy already reproduced by the direct approach. This shows that, although useful, transfer learning has to be used with care. What is confusing is that the variational energies appear to be stable. They follow almost a straight line, with only small differences between various versions of the RBM and even lattice size, as illustrated by the $N = 100$ data. Yet, these energies are approximately 2% higher than the energy of infinite DMRG (iDMRG) results in the expected PS phase, which were taken graphically from Ref. [37] and are marked by the orange triangles. However, the iDMRG results were obtained using a different type of lattice. Namely, an infinite cylinder with a circumference of 10 lattice points. Therefore, they are not directly comparable due to the finite-size effects. Nevertheless, the predicted position of the DS-PS transition point just below $J/J' = 0.69$ is too high and presents a conundrum.

Another, but related issue is the plaquette order parameter. Our results for the lattices with periodic boundary conditions suggest that there might be some fundamental problem with accessing the PS phase using RBM, because not all lattices show a significant PS order parameter where expected. However, this might be related to the problem of degeneracy of plaquette ordering. To shed more light on this problem, we tested other variants of the SSM lattices. In particular, a version where a perfect PS is expected and SSM with mixed boundary conditions that break the degeneracy.

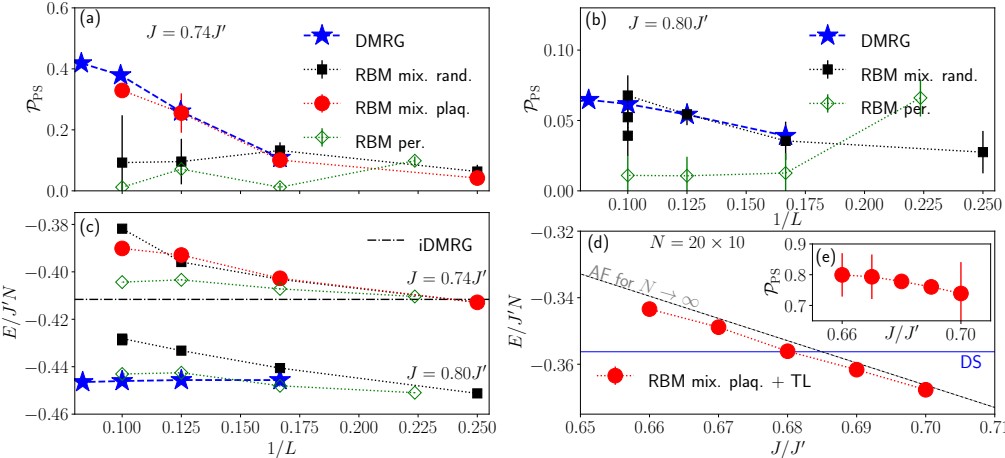

Figure 6: Comparison of finite size scaling of the order parameters for PS (a),(b) and variational energy (c) for $J/J' = 0.74$ and 0.8 between different methods and lattice boundary conditions. The empty green diamonds show the results for complex-valued RBM with $\alpha = 2$ for periodic boundary conditions ($L = \sqrt{N}$). The black squares and the red circles show the mixed boundary conditions ($L = \sqrt{N/2}$), where the former have been randomly initialized and the latter in the ideal PS ordering. Blue stars are DMRG results taken graphically from Ref. [38]. In panel (c), the upper half shows the $J/J' = 0.74$ results compared with the iDMRG result for an infinite cylinder with a circumference of $L = 10$ taken graphically from Ref. [37]. The bottom half compares our results for $J/J' = 0.8$ with the DMRG results from Ref. [38]. Panel (d) shows the evolution of the RBM variational energy as a function of $J/J'$ when initialized in a ideal PS and with transfer learning utilized in the learning process. The horizontal blue line shows the exact DS energy for $N = 20 \times 10$. The diagonal dashed gray line shows the asymptotic (large lattice) energy of AF ordering. Inset (e) shows the respective order parameter $\mathcal{P}_{PS}$.

**PS and mixed boundary conditions:** We performed a simple numerical experiment. We took the SSM lattice from Fig. 1 but set all interactions to zero except around the squares of type A. That is, we constructed a lattice of interacting spins on otherwise independent squares A. Starting with random initial conditions, MC with complex RBM and $\alpha = 2$ was able to converge and correctly capture the expected plaquette states on all accessible lattices. This means that there is no fundamental problem with PS ordering, and even a small RBM is expressive enough to describe this state. We then used these ideal plaquette states as an initial state for the MC calculations of the full SSM model at the respective lattices. Interestingly, for periodic boundary conditions, this did not lead to an improvement. PS ordering was strongly suppressed in the learning process, and we did not reach more favorable variational energies compared to those already obtained when starting from random initialization.

In accordance with, e.g., the recent work of Yang et al. [38], we decided to break the twofold degeneracy of expected PS ordering by changing periodic conditions to mixed ones. Following Ref. [38], we investigated cylinders with open boundary conditions in the $x$-direction with $L_x = 2L$ and periodic in the $y$-direction with $L_y = L$ so that $N = 2L^2$. In this geometry, the SSM has a preferred singlet plaquette pattern, and significant PS ordering is expected in the PS phase. We show in Appendix D that this ordering can be learned by a complex RBM with $\alpha = 2$ even for the lattice $N = 20 \times 10$. In addition, using this geometry also allowed us to compare our result directly with the DMRG results of Ref. [38]. Therefore, we first focus here on the parameters investigated there, although they are far away from the

DS-PS boundary and, therefore, show smaller $\mathcal{P}_{PS}$. In particular, we study $J/J' = 0.8$, for which we used random initial conditions, and $J/J' = 0.74$ where both the random and ideal plaquette states were used as initial states in the variational MC.

A comparison of the finite-size scaling of the PS order parameter and the variational energy obtained for periodic and mixed boundary conditions and different RBM strategies with the results of DMRG [38] (or iDMRG [37]) are shown in panels (a), (b), and (c) of Fig. 6. In general, periodic boundary conditions lead to lower variational energies, as demonstrated in Fig. 6(c), where the green diamonds closely follow the finite-size scaling predicted by DMRG results for $J/J' = 0.74$. RBM for lattices with mixed boundary conditions proved to be more difficult to train. On the other hand, they show a significant PS ordering. Although the RBM variational energy is generally larger than that of the DMRG and iDMRG studies, their PS order parameters are in reasonable agreement. However, here we draw attention to two observations. For $J/J' = 0.8$, where $\mathcal{P}_{PS}$ is low, a strategy with random initial conditions was sufficient to reproduce the DMRG results, as illustrated in Fig. 6(b). However, for $N = 20 \times 10$ three independent learnings lead to almost identical variational energies with difference smaller than 0.2% and therefore imperceptible in Fig. 6(b). Yet these states showed a noticeable difference in $\mathcal{P}_{PS}$ as visible in Fig. 6(b) (three black squares below each other). We attribute this problem to the combination of the overall low value of $\mathcal{P}_{PS}$ and its sensitivity to fluctuation of plaquete ordering between the squares of the lattice. The situation worsened for weaker coupling $J$. For $J/J' = 0.74$, learning with random initial states worked only for small lattices. For larger ones, the strategy where we initialized the RBM in an ideal PS gave much better results. The same number of iterations lead to lower energies and the expected $\mathcal{P}_{PS}$. This suggests that although the RBM with $\alpha = 2$ is capable of describing plaquette orderings, this state is difficult to learn without some help. Nevertheless, we utilized the strategy where an ideal plaquette state is used as an initial state to address another problem opened in the previous section.

We tested the position of the DS-PS phase transition point by focusing on $N = 20 \times 10$ with mixed boundary conditions. We started from the ideal plaquette ordering (see Fig. 10 in Appendix D) at $J/J' = 0.66$, therefore still in the expected DS phase, and then used transfer learning by sequentially increasing $J/J'$ for all points plotted in Fig. 6(d). Here, the blue horizontal line signals the exact energy of the dimer state. Although still slightly higher than the iDMRG result $J/J' = 0.675$, this lattice significantly reduced the estimate of $J$ up to which DS survives to $J/J' \approx 0.68$ compared to the above results with the periodic lattice $J/J' \approx 0.69$. Fig. 6(d) also shows that, in contrast to the periodic lattices, the PS ordering is robust here (see inset (e)). Actually, when artificially initialized, it can survive the learning process even for $J/J' < 0.68$ where the DS is the true ground state, although the learning rate plays an important role in this process. We used $\eta = 0.02$ ($\approx 300$ iterations) followed by $\eta = 0.003$ ($\approx$ (1000 iterations) at each step.

During our analysis, we have avoided the discussion of the possible SL and related DQCP which are compelling scenarios in part of a region here assigned to the PS. The reason is that due to several difficulties discussed above, e.g., the fact that our RBM results underestimate the PS order parameter even for $N = 20$ and large $\alpha$, a reliable analysis of SL and DQCP is currently beyond our reach. Nevertheless, here demonstrated expressiveness of a simple RBM with $\alpha = 2$ suggests that the problem can be indeed attacked by larger, more expressive, or specialized networks. A good candidate might be a composed GCNN that would combine networks for different characters of the symmetry group for particular lattice size and boundary conditions.

### 4.2.3 Magnetization plateaus

Historically, the most intriguing property of the SSM is its ability to describe fractional plateaus in magnetization as a function of an external magnetic field, which are also observed in real

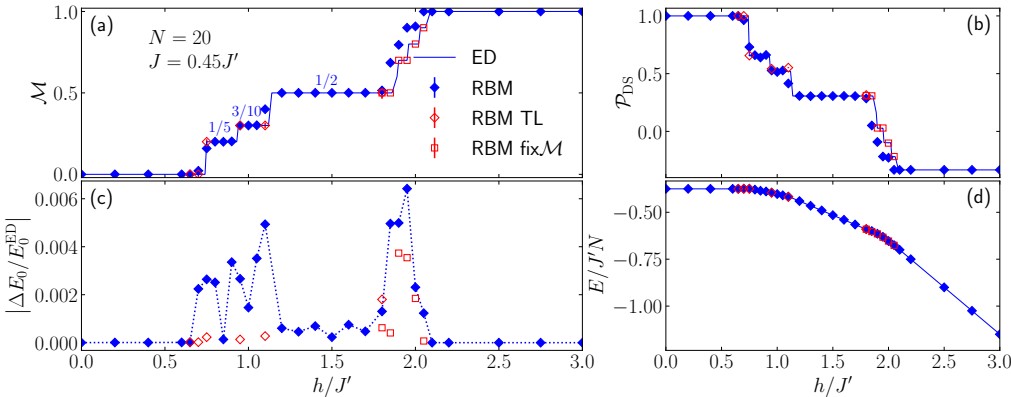

Figure 7: Comparison of ED (blue solid lines) and RBM with $\alpha = 2$ (symbols) results for $N = 20$ and $J/J' = 0.45$. Panels (a) and (b) show the magnetization and dimer state order parameter as functions of magnetic field. Panel (c) presents the relative error of the variational energy with respect to the ED result where blue dotted lines are just a guide to the eyes. Panel (d) shows the evolution of the normalized energy on external magnetic field. Blue filled diamonds represent the direct approach, empty red diamonds were obtained by utilizing the transfer learning discussed in the main text and the empty red squares by fixing $\mathcal{M}N$ to integer values from the vicinity of the direct approach.

materials. To address this problem through VMC, one has to drop the restriction of fixed $\mathcal{M} = 0$. In addition to significantly enlarging the Hilbert space, this also makes the optimization (learning) process a harder task. Moreover, each plateau represents a different ordering, and therefore, a challenge for NQS. However, as already demonstrated here, a simple RBM NQS with $\alpha = 2$ is sufficiently expressive to capture the main plateaus.

We assume only periodic boundary conditions and focus on the case $J/J' = 0.45$, which is inside the DS phase (at $h = 0$), where several broad plateaus are expected to form. The most stable ones, if allowed by the lattice size, should be the $\mathcal{M} = 1/2$ and $1/3$ plateaus [28, 42]. We start the discussion by benchmarking the RBM NQS results (blue filled diamonds in all panels of Fig. 7) against the ED results for the $N = 20$ lattice (blue solid lines). Clearly, the variational energy in panel (d) is in very good agreement with the exact one. The relative error plotted in panel (c) is much lower than 1% in the whole range of $h$. In addition, it shows a structure which can be understood by comparing the profile of the relative error dependence on $h$ with the normalized magnetization plotted in panel (a) and the DS order parameter in panel (b). Panel (a) shows that RBM NQS with $\alpha = 2$ is able to capture all main steps of the magnetization observed in the ED curve. The most stable are $\mathcal{M} = 0$, $1/2$ and $1$, followed by plateaus $1/5$ and $3/10$ that form in the range $0.7 \lesssim h/J' \lesssim 1.2$.

The stability of these plateaus is also reflected in the relative error. Although we do not use any restriction on $\mathcal{M}$, the relative error for $h/J' < 0.7$, where $\mathcal{M} = 0$, is negligible. In this region, the system stays in the DS ordering as revealed by panel (b). A similar situation exists for $h/J' \geq 2.1$. Here, the state is fully polarized ($\mathcal{M} = 1$) and, therefore, easy to reproduce with variational techniques. Other regions with very small errors in the variational energy are the central parts of the stable plateaus discussed above, as best illustrated by the $1/2$ one. Here RBM NQS gives a relative error below 0.1%. Consequently, the regions with the highest errors are related to the transitions between the stable plateaus. Here we also observe the largest deviations of the NQS magnetization (and $\mathcal{P}_{DS}$) from the ED results. These problematic regions can be divided into two types. The first one includes the step edges, i.e., the abrupt changes of the magnetization for $\mathcal{M} \leq 1/2$. The related convergence problems

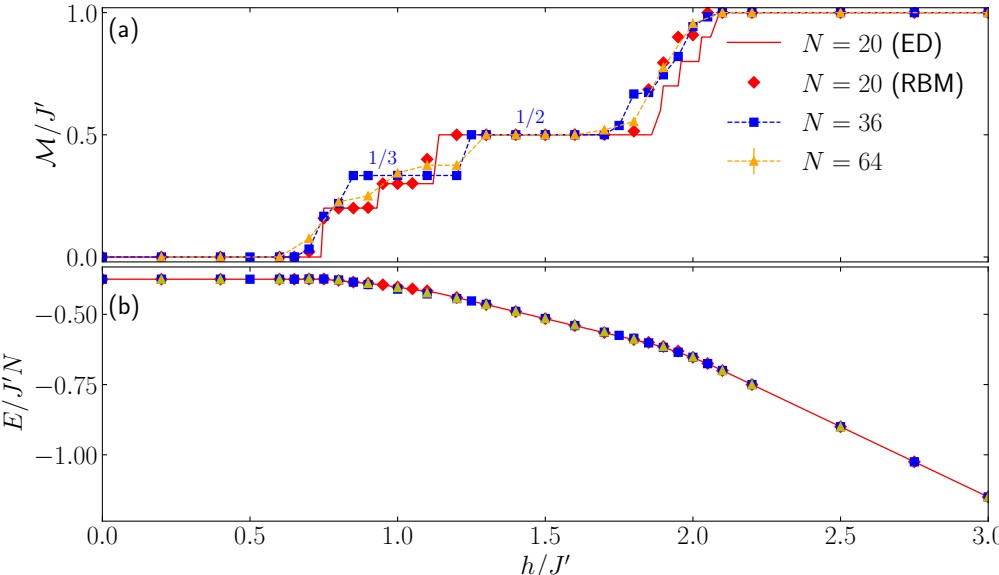

Figure 8: Comparison of the exact (red solid line) magnetization (a) and variational energy (b) results with VMC calculations utilizing RBM NQS with $\alpha = 2$ for lattices $N = 20, 36$ and $N = 64$ as functions of external magnetic field.

are similar to the difficulties of correctly capturing the precise position of the discontinuous phase transition discussed for $h = 0$ and $J/J' \approx 0.69$. As such, they can be also treated by the transfer learning. The red hollow diamonds in Fig. 7 were obtained by approaching the step edges from left and right using the RBM parameters learned in the centers of the neighboring plateaus as the initial input. Transfer learning clearly suppresses errors and gives the correct value of $\mathcal{M}$ even very close to discontinuities.

The second problematic region is at large magnetic field where the $\mathcal{M} = 1/2$ plateau transits into saturation $\mathcal{M} = 1$. It was shown only recently that this region can host exotic quantum states including several spin-supersolid phases [65]. Only one additional step-like rise of $\mathcal{M}$ from $1/2$ is expected here in the thermodynamic limit, which is followed by a continuous increase of magnetisation to $\mathcal{M} = 1$ as $h$ get larger. Nevertheless, the finite $N = 20$ lattice shows a number of very narrow transient steps in this region. This makes this region unsuitable for transfer learning, unless a much more refined grid of $h$'s is applied. On the other hand, the small lattice allowed us to test the actual expressiveness of RBM by fixing $\mathcal{M}N$ to integer values taken from the vicinity of the direct RBM results for $\mathcal{M}N$. The results with the lowest energies are depicted by the empty red squares, and they reproduce both $\mathcal{M}$ and the DS of the exact study. This proves that with correct learning strategy, RBM with $\alpha = 2$ is sufficient for the description of this rather complex evolution of the SSM ground state in the increasing magnetic field.

The stability of the magnetization plateaus must be confirmed on large lattices because the magnetization could be always discrete on finite clusters, yet continuous in the thermodynamic limit. Moreover, the lattice $N = 20$ is not divisible by three, so it cannot hold the important $1/3$ plateau. To show that RBM NQS can really capture these features, we address larger clusters. Fig. 8 presents, in addition to the exact (solid red line) and RBM (red diamonds) results for $N = 20$, the RBM results for $N = 36$ (blue squares) and $N = 64$ (yellow triangles). We stress here that these results were obtained with the direct approach. We have not used the transfer learning and fixed $\mathcal{M}$ to avoid the possibility that in this way we introduce a bias towards seemingly stable plateaus. Still, the results for $N = 36$ show stable flat steps in the magnetization which holds both the $1/2$ and the $1/3$ plateaus. Although the results for

$N = 64$ are less stable, they confirm the 1/2 plateau and clearly signal the formation of two additional plateaus for $h/J' < 1.2$. These are very encouraging results, as they again show that a simple RBM with small number of parameters is expressive enough to correctly capture the complicated magnetization dependence reflecting the underlying complex ordering of the quantum spins.

## 5 Conclusion

We have investigated the ground-state properties of the Shastry-Sutherland model via variational Monte Carlo with NQS variational functions. Our main goal was to show that a single and relatively simple NQS architecture can be used to approximate a wide range of regimes of this model. We have first tested and benchmarked several NQS architectures that are known from the literature to be suitable for different variations of the Heisenberg model. We discuss the role, advantages, and drawbacks of the NQSs that incorporate lattice symmetries and biases on the visible layer. We conclude that when precision, generality and computational costs are taken into account, a good choice for addressing larger SSM lattices without as well as with external magnetic field is a restricted Boltzmann machine NQS with complex parameters.

Focusing on RBM NQS allowed us to refine the learning strategy. We discovered that if a more precise MC sampling is used, then it is advantageous to run several (tens) short independent optimizations instead of a few long learnings. Using this strategy for the lattice $N = 20$ with periodic boundary conditions, we have demonstrated that already an RBM NQS with $\alpha = 2$ can accurately approximate the DS and AF phases and shows the onset of the PS ordering. It also gives a correct point of the discontinuous change of the DS regime to PS/AF. However, in its vicinity, there are the largest deviations from the exact results. Here, the variational energy can be significantly reduced by increasing $\alpha$, but this leads only to a small improvement in the estimation of the order parameters. Consequently, we used RBM NQS with $\alpha = 2$ to address larger lattices. Although reliable in the DS and AF phases in all lattices up to $N = 100$, PS proved to be more difficult to reach. However, this is partially a consequence of the degeneracy of the PS order. When broken by the usage of mixed boundary conditions, we were able to reproduce the DMRG result of Ref. [38] for the order DS parameter. Furthermore, we showed that the RBM NQS with $\alpha = 2$ is expressive enough to hold the PS order, although it might be difficult to train from a random initial state. To overcome this limitation, we introduced a strategy in which the RBM NQS is first trained on a lattice that enforces PS ordering and then this state is used as an initial state for the network in the relevant regime of SSM. This strategy allowed us to estimate the position of DS-PS phase transition to be $J/J' \approx 0.68$ for $N = 20 \times 10$, which is, taking into account the finite-size effects, in good accordance with the iDMRG result $J/J' = 0.675$. However, even when this strategy was used together with transfer learning, the training of RBM NQS for lattices with mixed boundary conditions proved to be more challenging than for periodic ones. For example, the finite-size scaling of the variational energy at $J/J' = 0.74$ closely follows the DMRG result; however, for mixed boundary conditions and $N = 20 \times 10$ the variational energy is still approximately 4% above the DMRG result [38].

A gradual increase of the magnetic field in SSM leads to formation of stable plateaus in the magnetization, each reflecting a different ground-state ordering. We have shown that RBM with $\alpha = 2$ can capture the relevant plateaus that form for the lattice sizes studied here. Transfer learning can then be utilized to refine the results.

To wrap it up, we have demonstrated that SSM is a good system for benchmarking NQSs and that a simple RBM NQS can be used to address its ground state in a broad range of regimes. This opens the possibility for NQSs to be used to address some unresolved questions related to

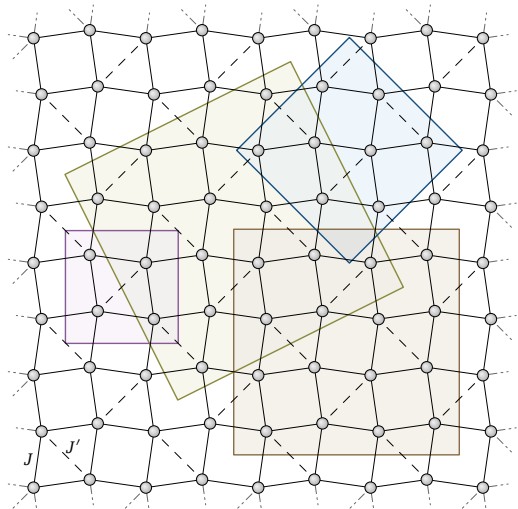

Figure 9: Shapes of tilted tiles of sizes $N = 4, 8, 16, 20$ used with periodic boundary conditions.

the SSM, e.g., the existence of the spin-liquid phase, DQCP and other exotic quantum phases expected in a finite magnetic field, or to precisely capture the size and character of additional steps in the magnetization for larger lattices. We, however, leave this for future more focused studies.

## Acknowledgments

We thank Artur Slobodeniuk for the helpful discussions and Alberto Marmodoro for helping us access additional computational resources.

**Funding information** This research was supported by the project e-INFRA CZ (ID:90140) of the Czech Ministry of Education, Youth and Sports (M.M., J.M.), P.B. acknowledges a support from the Czech Science Foundation via Project No. 19-28594X, and M.Ž. acknowledges a support from the Czech Science Foundation via Project No. 23-05263K.

## A  Lattice tiles

To benchmark various architectures we utilize ED. We use the Lanczos algorithm implemented in the `SciPy` library [66]. The only square (regular) systems tractable by this implementation, without extensive utilization of expected state symmetries, are of size 2×2 and 4×4. Therefore, we also constructed the so-called tilted regular-square clusters. They are depicted in figure 9 and each of them can be thought of as a single repeating building block of the infinite lattice. Clusters of sizes $N = 4, 8, 16, 20$ are accessible through ED and used to benchmark our NQS implementations (in the text we discuss only the results for $N \geq 16$).

## B  Visible biases in sRBM and pRBM

Here we show by contradiction that allowing uneven biases for sRBM is equivalent to constant biases when we enforce enough symmetries. Let us suppose that visible biases are kept non-constant $\left(a^f \to a_i^f\right)$ in the Eq. (9). We further assume the condition that $\forall i,j \; \exists g : g\sigma_i = \sigma_j$. This condition holds for every SSM tile.

It follows that $\sum_{g \in G} T_g(\boldsymbol{\sigma}^z)_i = C \sum_{i=1}^{N} \sigma_i^z = Cm^z$, where $C$ is the number of unique $g$ that fulfill the condition above. The first term in Eq. (9), after the generalization $a^f \to a_i^f$, can be rewritten as

$$\sum_{f=1}^{F} \sum_{g \in G} \sum_{i=1}^{N} a_i^f T_g(\boldsymbol{\sigma}^z)_i = \sum_{f=1}^{F} \sum_{i=1}^{N} a_i^f \sum_{g \in G} T_g(\boldsymbol{\sigma}^z)_i = Cm^z \sum_{f=1}^{F} \sum_{i=1}^{N} a_i^f = Cm^z \sum_{f=1}^{F} a^f . \qquad \text{(B.1)}$$

Thus, non-constant biases can be replaced by a constant value without loss of generality. Therefore, visible biases cannot be built into the sRBM as independent variational parameters.

On the other hand, pRBM is not limited in this way. This can be clearly seen after rewriting both ansätze into similar forms. First, the sRBM

$$\log \psi_{\boldsymbol{\theta}}(\boldsymbol{\sigma}^z) = \log \prod_{g \in G} \exp \sum_{f=1}^{F} \left\{ a^f \sum_{i=1}^{N} T_g(\boldsymbol{\sigma}^z)_i + \log\left[ 2\cosh\left( \sum_{i=1}^{N} w_i^f T_g(\boldsymbol{\sigma}^z)_i + b^f \right) \right] \right\},$$

and then pRBM

$$\log \psi_{\boldsymbol{\theta}}^G(\boldsymbol{\sigma}^z) = \log \sum_{g \in G} \chi_{g^{-1}} \exp \left\{ \sum_{i=1}^{N} a_i T_g(\boldsymbol{\sigma}^z)_i + \sum_{j=1}^{M} \log\left[ 2\cosh\left( \sum_{i=1}^{N} W_{ij} T_g(\boldsymbol{\sigma}^z)_i + b_j \right) \right] \right\}.$$

The sum (rather than the product) of exponentials makes it impossible to use an analogous reduction of visible biases as in Eq. (B.1). Note that the usage of visible biases does not typically lead to a significant increase of parameters ($+N$). Yet, they usually improve the convergence of the learning process for frustrated systems because they help to set the correct sign structure of the approximated state. Therefore, it is beneficial to include visible biases in the NQS parameters whenever possible.

## C  Symmetries

An infinite Shastry-Sutherland lattice has a *p4g* wallpaper group symmetry whose point group is $C_{4v}$ [67]. The character table of $C_{4v}$ is shown in Table 2. Each eigenstate of the SSM at infinite lattice must transform following one of the rows in the character table which, however, do not include the translations or glide reflections.

For finite lattices investigated in the paper, the table and the number of additional translations depend on the system size and shape (note that we are also using irregular lattices). Different small clusters can have different character tables with varying numbers of irreducible representations (irreps) [68, 69]. A detailed analysis of each lattice goes beyond the scope of our paper. In practical implementations, we used the automorphisms of the graph using routines implemented in NetKet [9, 56] and a particular line from its character table. For illustrative purposes, it is still useful to discuss the irreps of individual phases of the SSM on the infinite lattice.

Table 2: Character table of the $C4v$ point group describing symmetries of Shastry-Sutherland lattice.

|       | $E$ | $2C_4$ | $C_2$ | $2\sigma_v$ | $2\sigma_d$ |
|-------|-----|--------|-------|-------------|-------------|
| $A_1$ | 1   | 1      | 1     | 1           | 1           |
| $A_2$ | 1   | 1      | 1     | −1          | −1          |
| $B_1$ | 1   | −1     | 1     | 1           | −1          |
| $B_2$ | 1   | −1     | 1     | −1          | 1           |
| $E$   | 2   | 0      | −2    | 0           | 0           |

**DS**, described by Eq. (2), changes sign when we swap the spins in a dimer. More generally, the parity of the permutation determines the sign change. Consider a $L \times L$ square lattice, where $L$ is even, and a reflection symmetry along its diagonal axis ($\sigma_v$) within the squares containing the $J'$-bonds. The number of $J'$-bonds intersected by the axis is $L/2$ (considering the toroidal periodicity). For each of these bonds, a sign change occurs during the reflection while the sign of other dimers does not change. A similar argument can be constructed for the $C_4$ rotation. This has an important implication even for finite lattices. Namely, for regular lattices, the ground state of DS transforms under the trivial irrep $A_1$ if $L$ is divisible by 4, and under the antisymmetric irrep (corresponding to $B_2$) otherwise. This has some important consequences for the use of symmetries of some finite lattices, as discussed in the main text.

**PS** is twofold degenerate. Leaving out the translations, this means that it transforms under irrep E, which is the only irrep of dimension 2.

**AF** state analysis for finite lattices is rather complicated [68, 69]. If needed, we have assumed that AF transforms under trivial irrep $A_1$ (with and without the application of MSR).

## D   DS and PS in the RBM

**DS:** In principle, the complex-valued RBM is capable of representing a DS. For example, it can take advantage of visible biases (first term in Eq. (7)) and set them to reproduce the correct sign structure according to MSR. Since all nonzero weight coefficients have the same absolute value, the dense layer (second term in Eq. (7)) then needs only to identify these zero configurations and return a constant otherwise. An example of such construction is $b_j = 0$ and

$$W_{ij} = \begin{cases} i\frac{\pi}{2}, & \text{spin } i \in \text{dimer } j \,, \\ 0, & \text{otherwise} \,. \end{cases} \tag{D.1}$$

We number the dimers by index $j$ and $\cosh\left(\sum_i W_{ij}\sigma_i^z + b_j\right)$ is then one if the spins in dimer $j$ are antiparallel and zero otherwise. The size of the hidden layer corresponds to the number of dimers, specifically $N/2$, in this construction. By substituting Eq. (D.1) into Eq. (7), the DS state from Eq. (2) is reproduced. Notably, $W_{ij}$ nullifies all basis states that are not present in the DS state, while $a_i$ ensures the correct sign and $b_j$ can be adjusted to give a correct normalization. This shows that the RBM is, in theory, able to represent the DS state exactly. Whether, however, such a state can be learned, is in principle a different question. Nevertheless, the results in Fig. 8 clearly show that it can.

**PS:** A complex RBM with $\alpha = 2$ is expressive enough to encompass plaquette ordering even for large lattices. We demonstrate this for the $N = 20 \times 10$ lattice with mixed boundary

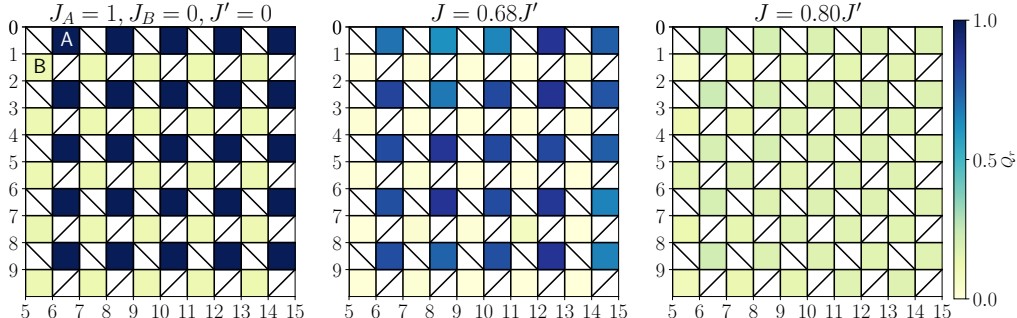

Figure 10: Plaquette ordering $Q_r$ in the central part of the SSM lattice $N = 20 \times 10$ with mixed boundary conditions. Here $Q_r = \langle \hat{Q}_r \rangle$ where $\hat{Q}_r = \frac{1}{2} \left( \hat{P}_r + \hat{P}_r^{-1} \right)$, with $\hat{P}_r$ being the cyclic permutation operator in square $r$. The left panel shows a toy model with $J_A = 1$ and $J_B = J' = 0$ where $J_A$ ($J_B$) is the coupling strength at the edges surrounding the squares of type A (B). The center and right panels show $Q_r$ for SSM with $J = 0.68$ (just below the DS-PS transition) and $J = 0.8$. The values of $Q_r$ for squares with diagonal bonds are not shown.

conditions (open in the $x$ direction and periodic in $y$). We start with a toy model, namely an SSM lattice with $J_A = 1$ and $J_B = J' = 0$, where $J_A$ ($J_B$) is the coupling strength at the edges surrounding the squares of type A (B), see Fig. 1. Starting from random initial state, the VMC converged to the plaquette ordering illustrated in the left panel of Fig. 10. We then use this state as an initial condition in the learning process for finite $J = J_A = J_B$ and $J' = 1$ in the range of values where plaquette ordering is expected. These results are shown in Fig. 6 and are discussed in the main text. In the central and right panels of Fig. 10 we show how the increase in $J$ suppresses the ordering of plaquettes in SSM.

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
