# Peer review of "Neural Network Quantum States analysis of the Shastry-Sutherland model"

_SciPost Physics, doi:SciPost Phys. Core 6, 088 (2023)_

## Round 1 · Referee Report · Anonymous (Referee 1) · 2023-5-22

Strengths

  • Benchmark of various restricted Boltzmann machine architectures as applied to the Shastry-Sutherland Heisenberg antiferromagnet.
  • Comparison of different options for implementing point group symmetries.

Weaknesses

  • The benchmark is not conclusive, the authors having opted for the least computationally demanding solution, rather than the "best" one in terms of accuracy.

Report

The authors study the phase diagram of the Shastry-Sutherland lattice using different variants of restricted Blotzmann machines (RBM) as a neural network quantum state. The goal is to find a single variational wavefunction that can correctly approximate the ground state in the entire phase diagram, at low computational cost. The authors conclude that a standard complex RBM gives the best trade-off between accuracy and computational cost and can capture the main features of magnetization plateaus.

Given that the number of numerical methods for frustrated spin systems are scarce, introducing and assessing a new variational ansatz is an important line of research. Especially in the Shastry-Sutherland lattice resolving small variational energy differences has led to new explanations of the nature of the magnetization plateaus. Therefore the authors' work opens up a promising direction, which justifies publication in SciPost Physics.

The most interesting aspect of the work, in my opinion, is the treatment of point group symmetries, which need to be chosen carefully, and the fact that taking into account the Marshal sign rule fails to give significant improvement, even in the AF phase, which is surprising.

However, there is a very basic observation:
In the benchmark in Table 1 and in the investigation of magnetization plateaus the authors choose the values J/J'=0.2 and J/J' = 0.45, respectively, which is inside of the DS phase (at h = 0). It should be pointed out that this parameter point is not only in the DS phase but that for J/J' < 0.5 the exact ground state is given by a product state of dimer singlets, according to Shastry and Sutherland's famous solution. If I am not mistaken, this should also apply to the N=16 and N=20 clusters in Table 1. Thus at J/J' = 0.2 some of the RBM architectures even struggle to represent an exact product state of dimers. Then the question boils down to how a product state of dimers is to be represented by an RBM.

Therefore, [1], the question of how to encode a dimer product state exactly as an RBM should be addressed, [2], the benchmark of different RBM architectures should be shown also in the dimer singlet phase for a parameter point 0.5 < J/J' < 0.68, i.e. where the ground state is not an exact product state of dimer singlets.

Provided that the above two criticisms are met, I recommend the paper for publication in Scipost Physics.

Finally, the NQS fail to quantitatively describe the order parameter in the plaquette singlet phase, which is two-fold degenerate. Did the authors try to remove the degeneracy by suitably chosen boundary conditions to alleviate this problem ?

Requested changes

See points [1] and [2] in the report.

  • The presentation of Fig. 6 could be improved by labeling the magnetization plateaus by their magnetization values.

  • validity: good
  • significance: good
  • originality: good
  • clarity: good
  • formatting: good
  • grammar: good

Author:  Martin Žonda  on 2023-11-03  [id 4091]

(in reply to Report 1 on 2023-05-22)

First of all we would like to thank the referee for the overall positive report as well as for the suggestions on how to improve our work. In the following we answer in detail all the raised questions and criticism and discuss how we have improved the manuscript by reflecting on them. (Original text of the referee to which we are reacting is repeated here for convenience and marked by /*** … ***/ )

/***
However, there is a very basic observation:
In the benchmark in Table 1 and in the investigation of magnetization plateaus the authors choose the values J/J'=0.2 and J/J' = 0.45, respectively, which is inside of the DS phase (at h = 0). It should be pointed out that this parameter point is not only in the DS phase but that for J/J' < 0.5 the exact ground state is given by a product state of dimer singlets, according to Shastry and Sutherland's famous solution. If I am not mistaken, this should also apply to the N=16 and N=20 clusters in Table 1. Thus at J/J' = 0.2 some of the RBM architectures even struggle to represent an exact product state of dimers. Then the question boils down to how a product state of dimers is to be represented by an RBM.

[1], the question of how to encode a dimer product state exactly as an RBM should be addressed
***/

The exact DS phase is always an eigenstate of the SSM and various theoretical studies suggest that it is the exact ground state up to J/J’=0.675(2) [for example, see PRX 9, 041037 (2019) or PRB 87, 115144 (2013)]. We consider this to be an exceptional quality of the model with respect to benchmarking new methods, such as NQS. It is straightforward to show that a complex-valued RBM with free biases on the input layer, i.e., the model used in the second part of our paper, is in principle capable of capturing such a state exactly. One possibility to represent a DS is to set the visible biases to reproduce the correct sign structure as discussed in the last paragraph of Sec. 3.2 and to set all the weights either to zero or a nonzero constant value. An example of such construction is now discussed in the new appendix D of the updated version of our paper. It shows that the RBM is in theory able to represent the DS state exactly. Learning the DS state is, of course, a different question.

Nevertheless, our benchmarks show that the DS can be learned [See Figs 4.(c), 5.(d) and 6(c)] even for L=100 (note that we have updated the data here) by using the second protocol. For small lattices even the first one is sufficient. We think that what is confusing here are these small but finite numbers in Table 1. They are not the lowest variation energies obtained. They are averages of the last 50 instances of the best runs. Because there are fluctuations in the variational energy due to the finite learning rate in the stochastic gradient descent as well as due to the Monte Carlo itself, these values are larger than the minimal one. We have opted for this, to have a protocol less sensitive to inaccuracies of the stochastic gradient descent method and, therefore, more reliable for the comparison of different architectures. However, the table does not necessarily show some hard limitations of the particular architecture. We have added the following note to the text, to clarify this to the readers: “Note that this means that we are not using just the lowest obtained energies but also test stability of the learning method. Consequently, the value of E50 is typically greater than zero even when the network is able to reproduce the state exactly.”

/***
[2], the benchmark of different RBM architectures should be shown also in the dimer singlet phase for a parameter point 0.5 < J/J' < 0.68, i.e. where the ground state is not an exact product state of dimer singlets.
***/

Previous studies indicate that, at least for here used finite lattices, the DS should be the ground state up to J/J’=0.675(2). Nevertheless, as also pointed out by another referee, the parameter range proposed by the referee is relevant because for example the J/J’ for SrCu2(BO3)2 at normal pressure is estimated to be 0.63. We have, therefore, added one column to Table 1, calculated for this value using the direct basis. These results are, with few exceptions, in a good qualitative agreement with the results of J/J’=0.2.

/***
Finally, the NQS fail to quantitatively describe the order parameter in the plaquette singlet phase, which is two-fold degenerate. Did the authors try to remove the degeneracy by suitably chosen boundary conditions to alleviate this problem ?
***/

We thank the referee for this question in particular. It is the main reason for our late reply as we have decided to investigate this problem in more detail. In addition, we pushed for significantly larger lattices. We have focused on the problematic region and found several interesting results. The most relevant ones we have included in the revised manuscript. First of all, already a RBM with alpha=2 is able to capture a fully developed plaquette singlet state. We tested it by using the Shastry-Sutherland lattice (graph) but setting all interactions to zero except the ones in squares where plaquettes are expected, i.e., we constructed a lattice of interacting spins on, otherwise, independent squares. RBM was able to correctly capture the expected plaquette states on all accessible lattices for both periodic (we artificially broke the degeneracy) and mixed boundary conditions. We then used these plaquette states as the initial state for the full model in the regime where the plaquette state is expected. Interestingly, for the periodic boundary conditions, this did not lead to an improvement. However, for the mixed ones it helped to lower the variational energy in the parameter regime where a significant plaquette order parameter is expected.

The introduction of mixed boundary conditions also allowed us to directly compare our results with DMRG results from Ref. [PRB 105, L060409 (2022)]. Admittedly, in the PS the RBM variational energy is typically larger than the one obtained by DMRG. Despite that, the obtained plaquette order parameters are in reasonable agreement. This again underscores the main message that just a simple NQS can capture qualitatively different orderings. The usage of mixed boundary conditions and larger lattices also lowered the estimated position of the phase transition point between the dimer state and the plaquette state to J/J’~0.68 and therefore much closer to the expected value of 0.675(2). The above points are now discussed in more detail in the main text of the manuscript as well as in the new appendix D. We have also added two multipanel figures [Fig. 6 and Fig. 10].

/***
The presentation of Fig. 6 could be improved by labeling the magnetization plateaus by their magnetization values.
***/

We have added the magnetization values to the main plateaus in the new figures 7 and 8 and also added the missing x axis label in the figure 8.

We have also focused on a thorough proofreading of our paper and corrected several other mistakes, typos, grammar mistakes as well as unclear statements. We think that all the corrections and the added results motivated by the questions of the referee led to a significant improvement of our manuscript.

---

## Round 1 · Referee Report · Anonymous (Referee 2) · 2023-5-23

Strengths

  1. A concise and clear review of the novel neural network quantum state approach

  2. A detailed examination of various neural network anatzes for the study of the Shastry-Sutherland model; establishing the conditions of the efficiency of the method.

Weaknesses

  1. A superiority of the method with respect to previous approaches has not been pointed out.

Report

The goal of the present paper to examine the complex physics of the Shastry-Sutherland model seems quite attractive and challenging. As it was reviewed in the manuscript the model exhibits quite a rich phase diagram and it is still an object of ongoing research. The application of the neural network quantum states can be a more efficient numerical approach for the frustrated model providing more accurate results for larger systems.

In the manuscript the authors broadly reviewed the variational Monte Carlo method and different architecture of neural network quantum states. In the original part, the authors confront the different versions of the neural network using the result for the systems of N=16, 20 spins. They concluded that NQS does not work well in the vicinity of the transition point between the dimer-singlet and plaquette-singlet phases. As a result, the transition point was not located precisely. The similar problem appears at the edge where the plaquette-singlet phase disappears. Finally, the results for the field-dependent characteristic have been presented.

The results given in the paper show some potential of the method, but also many questions regarding its applicability remain open, and it is not clear how the presented results outperform already known methods. I guess, the authors need to modify the conclusions in order to uncover the power of the method for the study of the diverse properties of the Shastry-Sutherland model.

Requested changes

  1. In the introduction, the authors review the Shastry-Sutherland model. I think it is not complete, and some other references can be added, e.g.

A. Abendschein and S. Capponi,  Phys. Rev. Lett. 101, 227201 (2008), where the exact diagonalization were performed for N=36 spins;

J. Dorier, K. P. Schmidt and F. Mila, Phys. Rev. Lett. 101, 250402 (2008) M. Nemec, G. R. Foltin and K. P. Schmidt, Phys. Rev. B 86, 174425 (2012), G. R. Foltin, S. R. Manmana and K. P. Schmidt, Phys. Rev. B 90, 104404 (2014), where the model has been studied by means of the perturbative continuous unitary transformations, i.e. perturbation theory.

  1. The definition of the PS order parameter given in Eq. (15) is not completely correct. I mean that it is barely possible to distinguish between the "singlet" and "empty" plaquettes used in the summation. Moreover, outside the PS phase this definition has no sense. I think that the authors need to reformulate it.

  2. In Fig. 5(c) the AF order parameter is given. However, this result is not discussed in the text. Is this result in accordance with the previous studies?

  3. The authors need to modify the conclusion in order to uncover the power of the method for the study of the diverse properties of the Shastry-Sutherland model.

  4. Please correct some typos There are misprints in the abbreviation: RMB ---> RBM in pages 9, 13 DF ---> DS in page 10 if if is suggested ---> if it is suggested in page 17

  • validity: high
  • significance: high
  • originality: high
  • clarity: high
  • formatting: excellent
  • grammar: excellent

Author:  Martin Žonda  on 2023-11-03  [id 4092]

(in reply to Report 2 on 2023-05-23)

First of all, we would like to thank the referee for the positive report as well as for pointing out some errors, missing references and for suggestions on how to improve our work. In the following, we answer in detail all the raised questions and criticism and discuss how we have improved the manuscript by reflecting on them. (Original text of the referee on which we are reacting is repeated for convenience.)

>*The results given in the paper show some potential of the method, but also many questions regarding its applicability remain open, and it is not clear how the presented results outperform already known methods. I guess, the authors need to modify the conclusions in order to uncover the power of the method for the study of the diverse properties of the Shastry-Sutherland model
>…
>4. The authors need to modify the conclusion in order to uncover the power of the method for the study of the diverse properties of the Shastry-Sutherland model.*

We have now added several direct comparisons of the NQS approach with DMRG and infinite DMRG (iDMRG) results. In particular, we have extended our analysis to larger lattices and results on lattices with mixed boundary conditions. We have focused on the regime where the ground state is expected to form the plaquette ordering. This refers to the region which is problematic already for small lattices. The reason is, that with improved learning we have been able to reproduce the exact results for the dimer phase even for large lattices. Which is now shown in our updated paper.

The comparison with DMRG in the plaquette phase showed that the finite-size scaling of the variational energy of RBM NQS for lattices with periodic boundary conditions is in a good agreement with that of DMRG. However, the learning for mixed boundary conditions proved to be much more challenging. The variational energy for large lattices (N>100) is higher than the DMRG result by several percent. Nevertheless, we were able to show that already a RBM NQS with alpha=2 is sufficiently expressive to describe the plaquette states. We now clearly state both the advantages, e.g., the fact that a very simple model without the usage of any symmetries can describe all main phases, and disadvantages, e.g. higher variational energy than predicted by (i)DMRG in the PS phase, of the approach in the conclusions.

>*1. In the introduction, the authors review the Shastry-Sutherland model. I think it is not complete, and some other references can be added, e.g.
>
>A. Abendschein and S. Capponi, Phys. Rev. Lett. 101, 227201 (2008),
>where the exact diagonalization were performed for N=36 spins;
>
>J. Dorier, K. P. Schmidt and F. Mila, Phys. Rev. Lett. 101, 250402 (2008)
>M. Nemec, G. R. Foltin and K. P. Schmidt, Phys. Rev. Lett. 101, 250402 (2008),
>G. R. Foltin, S. R. Manmana and K. P. Schmidt, Phys. Rev. B 90, 104404 (2014),
>where the model has been studied by means of the perturbative continuous unitary transformations, i.e. perturbation >theory.*

We thank the referee for turning our attention to additional literature. We have added all references to the relevant sentences in the introductory part.

>*2. The definition of the PS order parameter given in Eq. (15) is not completely correct. I mean that it is barely possible to distinguish between the "singlet" and "empty" plaquettes used in the summation. Moreover, outside the PS phase this definition has no sense. I think that the authors need to reformulate it.*

We thank the referee for pointing this out. We agree that the definition was incorrect and vague. We have corrected the formula in the updated version of the manuscript. Note that the new definition is written in a form that can be used also for uneven (Lx \neq Ly) lattices with mixed (periodic and open) boundary conditions.

>*3. In Fig. 5(c) the AF order parameter is given. However, this result is not discussed in the text. Is this result in accordance with the previous studies?*

The result is in qualitative agreement with previous works for example PRB 105(6), L060409 (2022). In the updated version of the manuscript we stress this by the following text:
Fig. 5 illustrates the usability of RBM for larger clusters. The presented results support the overall picture of the DS and AF phase separated by a narrow PS or at least its indication. Nevertheless, a much more thorough finite-size analysis would be necessary to assess the phase boundaries. For example, P_{AF} decreases with increasing system size in the whole relevant range of J′ which is in agreement with previous studies, e.g. [38]. Consequently, a careful and precise extrapolation of P_{AF} to the thermodynamic limit is needed to identify J above which the AF ordering prevails.

Please correct some typos
There are misprints in the abbreviation:
RMB .. RBM in pages 9, 13
DF .. DS in page 10
if if is suggested .. if it is suggested in page 17

We thank the referee for careful reading and for pointing out the typos. We have corrected all of them. In addition, we did a thorough proofreading of our manuscript and corrected several other mistakes, typos, grammar errors as well as unclear statements. Overall, the consideration and reflection on the questions and comments of the referee led to a significant improvement of our manuscript.

---

## Round 1 · Referee Report · Anonymous (Referee 3) · 2023-5-28

Strengths

Careful benchmark of neural-network quantum states on the example of the Shastry-Sutherland model.

Weaknesses

No new physics.

Report

The manuscript starts with a review of previous work and results, first of the Shastry-Sutherland model (section 2) and then Neural-network Quantum States (NQS) (section 3). The authors then benchmark NQS, specifically Restricted Boltzmann Machines (RBMs), first in sections 4.1, 4.2.1, and 4.2.2 for the zero-field ground state on lattices with $N\le 20$ sites. Finally, in section 4.2.3, the authors investigate the magnetization process for the example of $J/J'=0.45$.

Another Referee already pointed out that one might have cited further references. Even if I admit that there are so many investigations of the Shastry-Sutherland model that a complete overview is not possible, let me nevertheless mention the work by M. Jaime et al., PNAS 109, 12404 (2012) that one could have mentioned alongside Ref. [31].

While this is overall a nice benchmark with a large review component, I am a bit concerned about the novelty of the work. Specifically, the ED study Ref. [29] already investigated ground-state properties for significantly bigger systems than those investigated here, not to mention the iPEPS study Ref. [32]. Likewise, magnetization curves were already carefully investigated previously, e.g., in Refs. [31,33] and the PNAS by Jaime et al. It is true that here the authors make an effort to go to bigger lattices ($N\le 64$), but they focus on a parameter regime that is deeper in the dimer-singlet phase than the regime $J/J'\approx0.63$ relevant to SrCu$_2 $(BO$_3$)$_2$. Thus, there is no new physics. Even more, I understand that also the methods have been introduced elsewhere such that the only new element of the present work is the benchmark on a different model.

More on the level of detail, I believe that NQS and specifically RBM are reminiscent of tensor networks. Accordingly, I think that at least some comparison to previous tensor network and specifically iPEPS investigations would be appropriate, beyond just mentioning the existence of these works.

It might be useful to recall the acceptance criteria for SciPost Physics. According to https://scipost.org/SciPostPhys/about#criteria, a submission has to meet at least one of the following expectations in order "to be considered for publication in SciPost Physics: 1. Detail a groundbreaking theoretical/experimental/computational discovery; 2. Present a breakthrough on a previously-identified and long-standing research stumbling block; 3. Open a new pathway in an existing or a new research direction, with clear potential for multipronged follow-up work; 4. Provide a novel and synergetic link between different research areas." Given the preceding comments, I have a hard time seeing any of these criteria satisfied. The work is nevertheless a nice summary and comparative investigation of NQS, which should be suitable for publication in SciPost Physics Core after some minor revisions. I thus recommend that the authors submit a suitably revised version to SciPost Physics Core.

Requested changes

1- Add some comments on the comparison to tensor network methods, specifically iPEPS. 2- Complete the list of references, in particular add PNAS 109, 12404 (2012). 3- Make sure that all references have a DOI (concerns in particular Refs. [4,6,7,10,11,13,27,63,64]). 4- Fix formatting of titles. Ref. [28] presents one possibility to corrupt "SrCu$_2 $(BO$_3$)$_2$", Refs. [45-47] other versions, Refs. [39,40] corrupt "TmB$_4$", Ref. [16] has spurious lower-casing of "Gutzwiller", etc. I note that, e.g., Ref. [18] demonstrates that the authors know how to properly format references.

  • validity: high
  • significance: ok
  • originality: good
  • clarity: high
  • formatting: good
  • grammar: excellent

Author:  Martin Žonda  on 2023-11-03  [id 4093]

(in reply to Report 3 on 2023-05-28)

We would like to thank the referee for the constructive criticism as well as for the recommendation to publish our work as a regular paper in SciPost Physics Core. Nevertheless, we would still like to address all raised points in detail and discuss how we have improved our manuscript reflecting on the report. Hopefully, this will convince the referee that the revised version of our manuscript is worth publishing in SciPost Physics as well.

Another Referee already pointed out that one might have cited further references. Even if I admit that there are so many investigations of the Shastry-Sutherland model that a complete overview is not possible, let me nevertheless mention the work by M. Jaime et al., PNAS 109, 12404 (2012) that one could have mentioned alongside Ref. [31].

We have added the requested reference, as well others relevant referees to the overview part of the paper.

While this is overall a nice benchmark with a large review component, I am a bit concerned about the novelty of the work. Specifically, the ED study Ref. [29] already investigated ground-state properties for significantly bigger systems than those investigated here, not to mention the iPEPS study Ref. [32]. Likewise, magnetization curves were already carefully investigated previously, e.g., in Refs. [31,33] and the PNAS by Jaime et al. It is true that here the authors make an effort to go to bigger lattices (N≤64), but they focus on a parameter regime that is deeper in the dimer-singlet phase than the regime J/J′≈0.63 relevant to SrCu2(BO3)2. Thus, there is no new physics. Even more, I understand that also the methods have been introduced elsewhere such that the only new element of the present work is the benchmark on a different model.

We understand the concerns of the referee about the scientific novelty of the paper and, as discussed below, we have now made an effort to boost the paper in this respect. However, we would also like to point out to the two other possible route for publication in SciPost Physics listed on the webpage as well as in the report of the referee: Open a new pathway in an existing or a new research direction, with clear potential for multipronged follow-up work; Provide a novel and synergetic link between different research areas. We have shown in our paper that the Shastry-Sutherland model is not just another model on which the NQS approach can be applied. It is a model exceptionally suited for the task. This is partially because there are so many exact (even analytical) or close to exact (e.g., iDMRG) results obtained for this model as mentioned by the referee. But it is mainly because it has a complicated phase structure. There are continuous as well as discontinuous phase transitions, novel quantum phases, plateaus in the magnetizations, several open problems with conflicting results from different studies and the list goes on. We think that we are therefore opening a new pathway in existing research. We also combine two different research areas, namely the research of Shastry-Sutherland systems and NQS development.

The new elements are not just the benchmarks. First of all, we show that complicated networks are not necessarily the right answer to all problems. Then we introduced several strategies for using simple NQS on such a complicated system as the Shastry-Sutherland model. We have now demonstrated that a simple RBM can capture dimer state, AF state and also plaquette state, although the last one is difficult to learn. We discuss the advantages and disadvantages of transfer learning. We have also added some benchmarks for J/J′≈0.63 (see Table 1) relevant to SrCu2(BO3)2 at normal pressure. We have shown that NQS can deal with the step-like structure of the magnetisation plateaus. To our knowledge, this is a qualitatively novel result opening a new possible application of the method. For example, there are very recent results discussing a possibility of spin-supersolid orderings at finite magnetic fields [Nat Commun 14, 3769 (2023)] that could be now addressed with NQS. Further, we have now pushed the even larger lattices and also discussed mixed boundary conditions. We showed that this leads to a more realistic estimation of the position of the discontinuous phase transition, although the difficulties to learn PS prevail. To address this issue, we have introduced another strategy in the updated version of the manuscript to deal with this regime more efficiently. We also added comparison for periodic and mixed boundary conditions with DMRG and iDMRG results and discussed some problems this revealed.

More on the level of detail, I believe that NQS and specifically RBM are reminiscent of tensor networks. Accordingly, I think that at least some comparison to previous tensor network and specifically iPEPS investigations would be appropriate, beyond just mentioning the existence of these works.

There is definitely a connection between NQS and tensor networks and it was already investigated in the literature. See for example: D.-L. Deng, X. Li, and S. Das Sarma, Quantum entanglement in neural network states, Phys. Rev. X 7, 021021 (2017). J. Chen, S. Cheng, H. Xie, L. Wang, and T. Xiang, Equivalence of restricted boltzmann machines and tensor network states, Phys. Rev. B 97, 085104 (2018). Y. Levine, O. Sharir, N. Cohen, and A. Shashua, Quantum Entanglement in Deep Learning Architectures, Phys. Rev. Lett. 122, 065301 (2019). L. Pastori, R. Kaubruegger, and J. C. Budich, Generalized transfer matrix states from artificial neural networks, Phys. Rev. B 99, 165123 (2019). A. Borin and D. A. Abanin, Approximating power of machinelearning ansatz for quantum many-body states, Phys. Rev. B 101, 195141 (2020). C.-Y. Park and M. J. Kastoryano, Are neural quantum states good at solving non-stoquastic spin Hamiltonians? Phys. Rev. B 106, 134437 (2022) Or Sharir, A. Shashua, and G. Carleo, Neural tensor contractions and the expressive power of deep neural quantum states, Phys. Rev. B 106, 205136 (2022) Yet, if we understand the problem correctly, it is most probably the other way around. Let us cite the work of Sharirr at all. [PRB 106, 205136 (2022)] here: “ By directly constructing neural-network layers that perform tensor contractions, we show that efficiently contractible TNS can be constructed in terms of polynomially sized neural networks. Our result, in conjunction with previously established results on the entanglement capacity of NQS, then demonstrates that NQS constitute a very flexible classical representation of quantum states and that TNS commonly used in variational applications are strictly a subset of NQS.”

However, this does not mean that NQS are simply better, especially not when one focuses on relatively simple ones. Tensor networks states are a well-established general-purpose ansatz for quantum spin systems. The community has been using them for decades and, therefore, these techniques are already tuned to give the optimal results. On the other hand, NQSs are still a new approach. We are still searching for the optimal strategies as well as ideal networks. But there is potential. In this respect, one should also interpret the now added direct comparison of our results with the (i)DMRG results in the expected plaquette state phase. When focusing on this problematic region, the variational energy of the RBM is worse than the one given by ED (see N=20) or (i)DMRG (see the new results in Fig. 6). However, NQS can still give the correct order parameters, i.e., the physical structure of the ground state (new Fig. 6). Besides comparing the order parameter, we also show in Appendix D the structure of the ground-state clearly showing the plaquettes. The problem of too high variational energy, especially pronounced for large lattices, is related more to the learning process than to the expected limitations of small RBM NQS themselves.

Note that comparison in the DS is not necessary as RBM NQS reproduces the exact result as we discuss in the first paragraph of the added appendix D.

The work is nevertheless a nice summary and comparative investigation of NQS, which should be suitable for publication in SciPost Physics Core after some minor revisions. I thus recommend that the authors submit a suitably revised version to SciPost Physics Core. Requested changes 1- Add some comments on the comparison to tensor network methods, specifically iPEPS.

As discussed above, we have added several comparisons of the RBM NQS results with (i)DMRG. In particular, please see Fig. 6 and the detailed discussion related to it. Some comparison (not discussed above) is also shown in Fig. 5b, however, there we compare data for infinite cylinder lattices to small periodic ones. Therefore, this example is not authoritative for the overall assessment of the NQS results. We have also expanded the paper by additional appendix D where we demonstrate that RBM is expressive enough to encompass both dimer and plaquette ordering.

2- Complete the list of references, in particular add PNAS 109, 12404 (2012). 3- Make sure that all references have a DOI (concerns in particular Refs. [4,6,7,10,11,13,27,63,64]). 4- Fix formatting of titles. Ref. [28] presents one possibility to corrupt "SrCu2 (BO3)2", Refs. [45-47] other versions, Refs. [39,40] corrupt "TmB4 ", Ref. [16] has spurious lower-casing of "Gutzwiller", etc. I note that, e.g., Ref. [18] demonstrates that the authors know how to properly format references.

We have put a significant effort into adding missing references and correcting wrong formats as well as spurious lower-casings. In addition, we have provided a thorough proofreading, corrected some typos, grammar mistakes, inconsistencies in referencing equations and figures and reformulated some sentences to make the text clearer.

We again thank the referee for the overall kind assessment of our work and recommendation for publication in SciPost Physics Core. Nevertheless, to answer the criticism of the referee, we have now significantly improved the paper. We have also added benchmarks, new strategies and new results. Therefore, and with respect to the recommendation of the two other referees, we kindly ask the referee to reconsider again the suitability of our work for the SciPost Physics.

---

## Round 2 · Referee Report · Anonymous (Referee 2) · 2023-11-16

Report

In the revised version the authors took into account referees' remarks. They increased the clarity of the manuscript. The authors provided quite an extensive overview of the neural network methods for quantum systems and then employed different schemes of them to study some ground-state phases of the Shastry-Sutherland model. The authors examine in detail the precision of the obtained results when changing the number of layers in the neural network (NN). One of the main messages is that a shallow NN can be sufficient to recover some phases of the model.

Unfortunately, the main question of the previous reviewer reports "what is the advantage of the methods employed in the manuscript with respect to other numerical methods" is not well elucidated. I understand that the current NQS methods could be computationally less demanding with respect to other methods, but on the other hand the results presented here concern the topics more or less well understood in the Shastry-Sutherland model. Therefore, no new knowledge of such a complex problem, as the Shastry-Sutherland model, has been gained and the current study does not meet the acceptance criteria for SciPost Physics.

Nevertheless, it is a nice paper which can be published in SciPost Physics Core.

Requested changes

Below Eq.(14):
The Planck constant is redundant in (-3/4)\hbar^2. The spin operators below Eq.(1) has been introduced without it.

---

## Round 2 · Referee Report · Anonymous (Referee 1) · 2023-11-18

Strengths

The work opens up a new research direction and gives important guidelines for further improvements on simulating Shastry-Sutherland model with different architectures of neural network quantum states.

Weaknesses

No new results on the Shastry-Sutherland model are reported.

Report

The authors have answered all my question in great detail. In particular,
the representation of an exact dimer solid state has been addressed, and it has been argued that there may be issues in learning this state using the variational principle even though it is in the variational scope of the ansatz.
The authors have also tested how different types of boundary conditions that break the ground state degeneracy
affect the results in the plaquette singlet phase.

Given these improvements of the manuscript, I recommend publication in SciPost Physics
as the paper opens up a new research avenue by bridging the field of
neural quantum states with topical research on the
Shastry-Sutherland model, which is of great current interest due to direct experimental
realizations and which is hard to simulate numerically as only few methods can tackle this frustrated system.
One can expect that the authors' work will inspire and accelerate further studies.

---

## Round 2 · Referee Report · Anonymous (Referee 3) · 2023-11-25

Report

I do appreciate the effort that the authors have put into revising their manuscript, in particular the additional results for $J/J'=0.63$ in Table 1, the new Appendix D, and the new figures 6 and 10. Nevertheless, there is still no new physics in the manuscript and from a technical point of view the present work relies on the NetKet libraries [9,56].

True, DMRG and iPEPS have a longer development history than neural network quantum states, but the present results are not only less accurate, but I do not see a major advance in the available toolbox either. Even if the present benchmarks remain interesting, this is in my opinion a clear case for SciPost Physics Core.

Requested changes

When rereading the manuscript, I noted a few details that the authors might wish to correct on the proofs:
1- I believe that the work that defined the name "Shastry-Sutherland model" should be cited in the first paragraph of the Introduction, and not just as Ref. [55] on page 5.
2- H. Kageyama et al., Phys. Rev. Lett. 82, 3168 (1999) is a central paper on SrCu$_2$(BO$_3$)$_2$ that has so far been overlooked. I recommend to cite it alongside / before Ref. [44].
3- There are three "dimmer"s in the manuscript (on pages 5, 19, and 25).
4- First paragraph of page 16: I believe that the correct mathematical term is "monotonically" (not "monotonously").
5- Top of page 23: the "differed ground state ordering" should probably read "different ground state ordering".
6- Ref. [59] seems to be published in PMLR 48:2990-2999, 2016 (albeit no DOI).

---

## Round 2 · Author Response

Dear Editor,

We hereby resubmit our manuscript for publication in SciPost Physics. The previous version was reviewed by three referees with an overall positive assessment of our work. Nevertheless, they also raised a number of questions, constructive criticism and useful recommendations. By reflecting on all reports of the referees and with the aim to further improve our manuscript we have prepared a new and extended version of the manuscript.

All changes are discussed in detail in the answers to the referees.

On behalf of all authors,
Martin Žonda

---

## Round 2 · List of Changes

By reflecting on all the questions, criticism and suggestions raised by the referees and with the aim to improve our manuscript we have made the following changes:

We have added benchmarks for J/J’ = 0.63 to Table 1 and briefly discuss these results in the main text.

We were able to converge the results for L=100 and, therefore, corrected the related data in the figure 5 and simplified the relevant text.

We have added iDMRG results taken graphically from the reference Phys. Rev. X 9(4), 041037 (2019) to the panel (e) in figure 5 and discuss the difference in the main text.

We have significantly expanded section 4.2.2 by an investigation of the plaquette phase for lattices with mixed boundary conditions (open and periodic). We have added a new composite figure 5. It shows a comparison of the RBM NQS results to DMRG taken from reference Phys. Rev. B 105(6), L060409 (2022) as well as the investigation of the position of the discontinuous phase transition between the dimer and plaquette phase. The results are discussed in length in the part “PS and mixed boundary conditions“. There we also address the strengths and weaknesses of the RBM NQS when compared to DMRG results for the SSM.

We have added a new appendix D in which we first show how RBM can encode dimer state and then demonstrate that it is expressive enough to encompass the plaquette ordering as well.

We have updated the conclusions to include the new results.

We have expanded the list of relevant references by works suggested by referees as well as by a few newly published works on the topic.

We corrected the wrong formats as well as spurious lower-casings in the list of references.

We have corrected typos, grammar mistakes, inconsistencies in referencing equations and figures and reformulated and simplified some sentences to make the text clearer.

A more detailed description of the changes can be found in our response to the referees.

---

## Editorial Decision

published